# Stochasticity in the miR-9/Hes1 oscillatory network can account for clonal heterogeneity in the timing of differentiation

**Nick E Phillips[1], Cerys S Manning[1†], Tom Pettini[1†], Veronica Biga[1†], Elli Marinopoulou[1†], Peter Stanley[1†], James Boyd[1], James Bagnall[1], Pawel Paszek[1], David G Spiller[1], Michael RH White[1], Marc Goodfellow[2,3,4], Tobias Galla[5], Magnus Rattray[1], Nancy Papalopulu[1]\***

[1]Faculty of Biology, Medicine and Health, University of Manchester, Manchester, United Kingdom; [2]College of Engineering, Mathematics and Physical Sciences, University of Exeter, Exeter, United Kingdom; [3]Centre for Biomedical Modelling and Analysis, University of Exeter, Exeter, United Kingdom; [4]EPSRC Centre for Predictive Modelling in Healthcare, University of Exeter, Exeter, United Kingdom; [5]Theoretical Physics, School of Physics and Astronomy, University of Manchester, Manchester, United Kingdom

**Abstract** Recent studies suggest that cells make stochastic choices with respect to differentiation or division. However, the molecular mechanism underlying such stochasticity is unknown. We previously proposed that the timing of vertebrate neuronal differentiation is regulated by molecular oscillations of a transcriptional repressor, HES1, tuned by a post-transcriptional repressor, miR-9. Here, we computationally model the effects of intrinsic noise on the *Hes1*/miR-9 oscillator as a consequence of low molecular numbers of interacting species, determined experimentally. We report that increased stochasticity spreads the timing of differentiation in a population, such that initially equivalent cells differentiate over a period of time. Surprisingly, inherent stochasticity also increases the robustness of the progenitor state and lessens the impact of unequal, random distribution of molecules at cell division on the temporal spread of differentiation at the population level. This advantageous use of biological noise contrasts with the view that noise needs to be counteracted.

*For correspondence: nancy.papalopulu@manchester.ac.uk

†These authors contributed equally to this work

Competing interests: The authors declare that no competing interests exist.

## Introduction

The nervous system is a complex organ system with many different cell types. These are generated by actively dividing progenitor cells in a sequential order during embryogenesis, while progenitor cells themselves change their character along the way (*Götz and Huttner, 2005*). In cases where histogenesis of neural tissues has been examined, such as the mammalian cortex and the vertebrate retina, a temporal order has been found such that certain cell types are generated early while others are generated late (*Franco et al., 2012*; *Bertrand et al., 2002*; *Boije et al., 2014*; *Greig et al., 2013*; *Franco and Müller, 2013*; *Goetz et al., 2014*; *Shen et al., 2006*; *Gaspard et al., 2008*). In order to understand neural development it is essential to understand how the timing of differentiation is regulated.

The predominant view is that the timing of differentiation is precise and tightly controlled. Indeed, given that the final developed nervous system is consistently structurally organised, this is an

intuitive assumption. Therefore, it was surprising that recent experiments based on single-cell lineage mapping in the zebrafish retina have shown that the timing of differentiation can be a fundamentally stochastic process, although the collective behaviour of many progenitors forms a reproducible retina (*He et al., 2012*). A similar conclusion was reached in a study where rat retinal progenitor cells were cultured at single-cell density and constant extracellular environment (*Gomes et al., 2011*). In addition to the retina, lineage tracing has also demonstrated that progenitors produce a temporally ordered but variable number of neurons within the mouse neocortex (*Shen et al., 2006*; *Gaspard et al., 2008*) although other studies reported a high degree of determinism in this system (*Gao et al., 2014*). Nevertheless, inherent stochasticity may be a widespread phenomenon in differentiation.

Previous models aiming to explain this phenomenon, such as the branching models of population dynamics (*Rué and Martinez Arias, 2015*; *Klein and Simons, 2011*), take into account 3 types of division: symmetric proliferative, asymmetric and symmetric differentiative. The probabilities of division type are either fixed (e.g. in adult tissue homeostasis; reviewed in Simons and Clevers (*Simons and Clevers, 2011*)) or change with time (e.g. in developing tissues (*Slater et al., 2009*)), but are usually fitted to the data (*He et al., 2012*; *Gomes et al., 2011*). Such models are useful in understanding population dynamics but lack a molecular mechanistic explanation.

At a molecular level, the balance between differentiation and the maintenance of progenitors in the developing nervous system is regulated by transcriptional repressors such as the *Hes* gene family, including *Hes1* and *Hes5* (*Kageyama et al., 2007*, *2008*). When *Hes* genes are absent, neural progenitors prematurely differentiate and cause a wide range of defects in brain formation (*Hatakeyama et al., 2004*; *Nakamura et al., 2000*). Conversely, overexpression of *Hes* genes leads to inhibition of neurogenesis and over-maintenance of neural progenitors (*Ishibashi et al., 1995*).

The development of live-cell imaging with unstable Luciferase (LUC) reporters has shown that the dynamics of *Hes1* gene expression changes during neural development (*Imayoshi et al., 2013*). *Hes1* is expressed in an oscillatory manner with a period of around 2 hr in neural progenitors (*Imayoshi et al., 2013*; *Shimojo et al., 2008*) but is expressed at a low steady state in differentiated neurons (*Imayoshi et al., 2013*; *Sasai et al., 1992*). Based on expression dynamics and the functional studies mentioned above, it has been proposed that a *Hes1* oscillatory state is necessary for the maintenance of progenitors, while low, non-oscillatory levels are associated with a transition to neuronal differentiation (*Kageyama et al., 2008*). The most direct evidence for the functional importance of oscillatory dynamics in general, comes from optogenetics studies of the *Hes1* targets *Ascl1* (*Imayoshi et al., 2013*) and *Dll1* (*Shimojo et al., 2016*). It was shown that light-induced oscillatory expression of *Ascl1* increased the proportion of dividing cells in *Ascl1*-null neural progenitors, whereas sustained *Ascl1* expression increased the efficiency of neuronal differentiation (*Imayoshi et al., 2013*). It has similarly been shown that light-induced sustained expression of the Delta ligand *Dll1* leads to higher levels of the cell cycle inhibitor p21 than oscillatory expression (*Shimojo et al., 2016*). Together, these suggested that the expression dynamics of *Ascl1* and *Dll1* encode information for a choice between proliferation and differentiation within neural progenitors. The hypothesis that gene expression dynamics change as cells make cell-state transitions in development is consistent with previous studies in theoretical biology (*Furusawa and Kaneko, 2012*; *Huang, 2011*; *Garcia-Ojalvo and Martinez Arias, 2012*; *Rué and Martinez Arias, 2015*). In these studies, the interactions of multiple genes in regulatory networks can lead to the emergence of transient stem cell dynamics, which evolve to an attracting stable configuration of gene expression corresponding to distinct cell types.

Experimental and theoretical work has shown that *Hes1* oscillates in neural progenitors possibly due to a combination of delayed negative self-repression and relatively fast degradation of *Hes1* mRNA and HES1 protein, previously measured in fibroblasts (*Jensen et al., 2003*; *Monk, 2003*; *Hirata et al., 2002*; *Momiji and Monk, 2008*). Until recently it was not understood how oscillations of *Hes1* could be terminated and the timing of differentiation controlled. Recent experimental results have shown that *Hes1* is a primary target of the microRNA miR-9, and HES1 also periodically represses the transcription of miR-9, thus forming a double negative feedback loop (*Bonev et al., 2012*). However, mature miR-9 is very stable and accumulates over time in a gradual manner. It has been proposed that accumulating levels of miR-9 beyond a certain level can cause oscillations of *Hes1* to cease, leading to differentiation (*Bonev et al., 2012*; *Tan et al., 2012*). Experimentally it has been shown that *Hes1* is a target of miR-9, that depleting miR-9 prevents or delays differentiation

and that *Hes1* changes dynamics of expression as cells differentiate (*Bonev et al., 2012*; *Bonev et al., 2011*; *Imayoshi et al., 2013*; *Tan et al., 2012*; *Coolen et al., 2012*). However, a theoretical approach unifying these phenomena was lacking.

The *Hes1*/miR-9 interaction has been mathematically modelled using deterministic delay differential equations (*Goodfellow et al., 2014*). This showed that interactions within the *Hes1*/miR-9 network can lead to the emergence of bistability (alternative cell fates) and Hopf bifurcations (separating stable states from an oscillatory regime). Computationally it was shown that indeed, accumulation of miR-9 can lead to a transition from oscillatory to stable expression of HES1 protein and that the timing of this transition is controlled by the parameters and the initial conditions of the model. Although this mathematical model provided a useful starting point, a deterministic system is not able to explain the stochastic nature of the differentiation process that has been reported in vivo.

Here, we show that high cell-to-cell variability exists in the abundance of *Hes1* mRNA and protein in undifferentiated neural cells (progenitor and stem cells) as evidenced by quantitative single-molecule Fluorescent in Situ Hybridisation (smFISH) and Fluorescence Correlation Spectroscopy (FCS), respectively. Furthermore, the copy number of *Hes1* mRNA, protein and miR-9 per cell is low, (determined by smFISH, FCS and stem loop qRT-PCR, respectively) supporting the inclusion of 'finite number' intrinsic stochasticity in the computational model of the *Hes1*/miR-9 interaction. Oscillations produced by a stochastic model match single-cell imaging data well and lead to systematic changes in the dynamics and output of this network. In the deterministic model, equivalent cells (i.e clonal or isogenic) that start from the same initial conditions terminate oscillations i.e. 'differentiate' at exactly the same time. However, in the stochastic framework the time-to-differentiation of such cells becomes distributed, with the spread of the distribution inversely related to the absolute number of interacting molecules. Due to the ability of intrinsic noise to induce oscillatory dynamics, *Hes1* oscillations are maintained for a greater range of parameters than in the deterministic system, which can be interpreted as increased robustness of the progenitor state in the stochastic model. Computationally, the average time of differentiation can be shifted by varying the molecular copy number or by varying the initial miR-9 levels. This supports the idea that the number of interacting molecules within a cell is an important parameter that can be tuned to control the onset of differentiation. We validate predictions of our stochastic model by measuring the time distribution of differentiation when miR-9 is added to the system. Finally, we show that the stochastic network is able to better 'absorb' noise attributable to unequal distribution of some molecular components at division, maintaining its output as if the divisions were symmetric and thus giving a robust outcome at a population level.

Together, our work demonstrates that randomness originating in the finite copy number within transcriptional programmes provides a theoretical framework to understand the heterogeneity observed in single-cell time series of neural stem (NS) cells and its consequences for neural development. Developing systems may exploit stochasticity in gene regulatory networks to stagger differentiation time, cope with noise introduced by unequal inheritance of components and safeguard the progenitor state.

## Results

### Developing a stochastic model for the *Hes1*/miR-9 oscillator

Previous analysis focussed on a model of the *Hes1*/miR-9 network using delay differential equations (*Goodfellow et al., 2014*), which are themselves an extension of earlier models by *Jensen et al. (2003)* and *Monk (2003)*; *Momiji and Monk (2008)*, as outlined in the Materials and methods.

A deterministic system relies on differential equations, making an assumption that variables such as mRNA and protein can take a continuous range. This is very familiar to the biologist and such equations are commonly used for modelling biological processes (*Alon, 2007*). However, in reality, biochemical reactions are the result of random encounters between discrete numbers of molecules, and some of these molecules may be present at low numbers. The finite number of molecules interacting within the system leads to inherent randomness known as intrinsic stochasticity (*Swain et al., 2002*).

Thus, while the units of the deterministic differential equations are concentrations, a stochastic model requires a description in terms of discrete numbers of molecules. To convert from concentration into the number of molecules we use the system size parameter, $\Omega$, which can be seen as a measure of the volume of the biological system (*Kaern et al., 2005*; *Elf and Ehrenberg, 2003*). Concentrations and copy numbers (i.e. number of molecules) are then related via

$$\text{Copy number} = \Omega \times \text{Concentration} \qquad (1)$$

The copy number can increase when the system size (volume) increases and the concentration is constant, or when the system size (volume) is constant and the concentration increases.

In order to investigate the effects of intrinsic noise on the dynamics of differentiation we reformulated the *Hes1*/miR-9 model in a stochastic framework, which describes the system in terms of its underlying microscopic reactions, namely the production and degradation of *Hes1* mRNA, HES1 protein and microRNA miR-9. We use three complementary stochastic modelling approaches: simulation of the Chemical Master Equation (CME) with the Stochastic Simulation Algorithm with delay (dSSA), simulation of the Chemical Langevin Equation (CLE) and the Linear Noise Approximation (LNA) as described fully in the Materials and methods.

Briefly, the CME uses the network of interactions and their associated rates to formulate how each reaction acts to increase or decrease the likelihood of transitioning to a new state. For example, if the number of protein molecules is very high, the next most likely reaction may be a protein degradation event. Other reactions can still be triggered, however, and this leads to the heterogeneity over time.

Single realisations of the CME were simulated computationally using Gillespie's Stochastic Simulation Algorithm with delay (dSSA) (*Gillespie, 1977*; *Anderson, 2007*; *Cai, 2007*) which is a statistically exact solution of the master equation but can be computationally expensive, particularly at high system size ($\Omega$) because every reaction is simulated. Therefore, to investigate different parameters and to understand the behaviour of a population of cells, both of which require a large number of repeats, we used methods that are more computationally efficient such as Chemical Langevin Equations (CLE) and the Linear Noise Approximation (LNA), which add a random fluctuating term by bundling reactions together (*Gillespie, 2000*; *Grima et al., 2011*). The Linear Noise Approximation (LNA) is a (semi-)analytical approach that does not require multiple simulations and is hence the fastest of all the methods for scanning parameter space. However, it can be less accurate at low system size (*Grima, 2012*).

## A 'finite number' stochastically simulated network is justified by experimental data

In our conceptual framework, the timing of differentiation corresponds to the time required to reach a low stable HES1 state in the model, brought about by an increase in miR-9 over time (*Goodfellow et al., 2014*). The increase of miR-9 over time in cultured neural progenitor cells has been shown before (*Bonev et al., 2012*). Here, using a miR-9 sensor we show that the activity of miR-9 increases specifically in neuronally differentiated cells (*Figure 1—figure supplement 1*).

In order to justify the finite-number approach in our modelling we used smFISH to determine the absolute copy number of *Hes1* mRNA in C17.2 neural progenitor cells and neural stem cells (NS) expressing GFP from the *Mapt* (*tau*) locus (Tau-GFP NS cells). Multiple fluorescent probes were designed to target the *Hes1* transcript (*Figure 1A*) and the accuracy of smFISH for absolute quantitation was validated in *Figure 1—figure supplement 2*. The *Hes1* mRNA copy number was highly variable between individual cells grown under identical conditions and also between cell lines (*Figure 1B and D*), but generally quite low. In C17.2 neural progenitor cells, we detected an average of 72 copies *Hes1* mRNA/cell by smFISH (*Figure 1C*), and an average of 88 copies by qRT-PCR (data not shown), confirming the result. In Tau-GFP NS cells grown under proliferative conditions (GFP negative cells; *Figure 1D and E*), the average was only 11 copies of *Hes1* mRNA/cell by smFISH. The long-tail distribution of mRNA molecules in NS cells may be indicative of slow switching between transcriptionally active and inactive promoter states (*Munsky et al., 2012*). Other studies using smFISH have reported numbers of mRNA molecules ranging from a few copies to $10^3$ (*Neuert et al., 2013*; *Bahar Halpern et al., 2015*; *Battich et al., 2015*). A few spontaneously differentiated cells, identified by high-intensity GFP expression (*Figure 1—figure supplement 3*), were

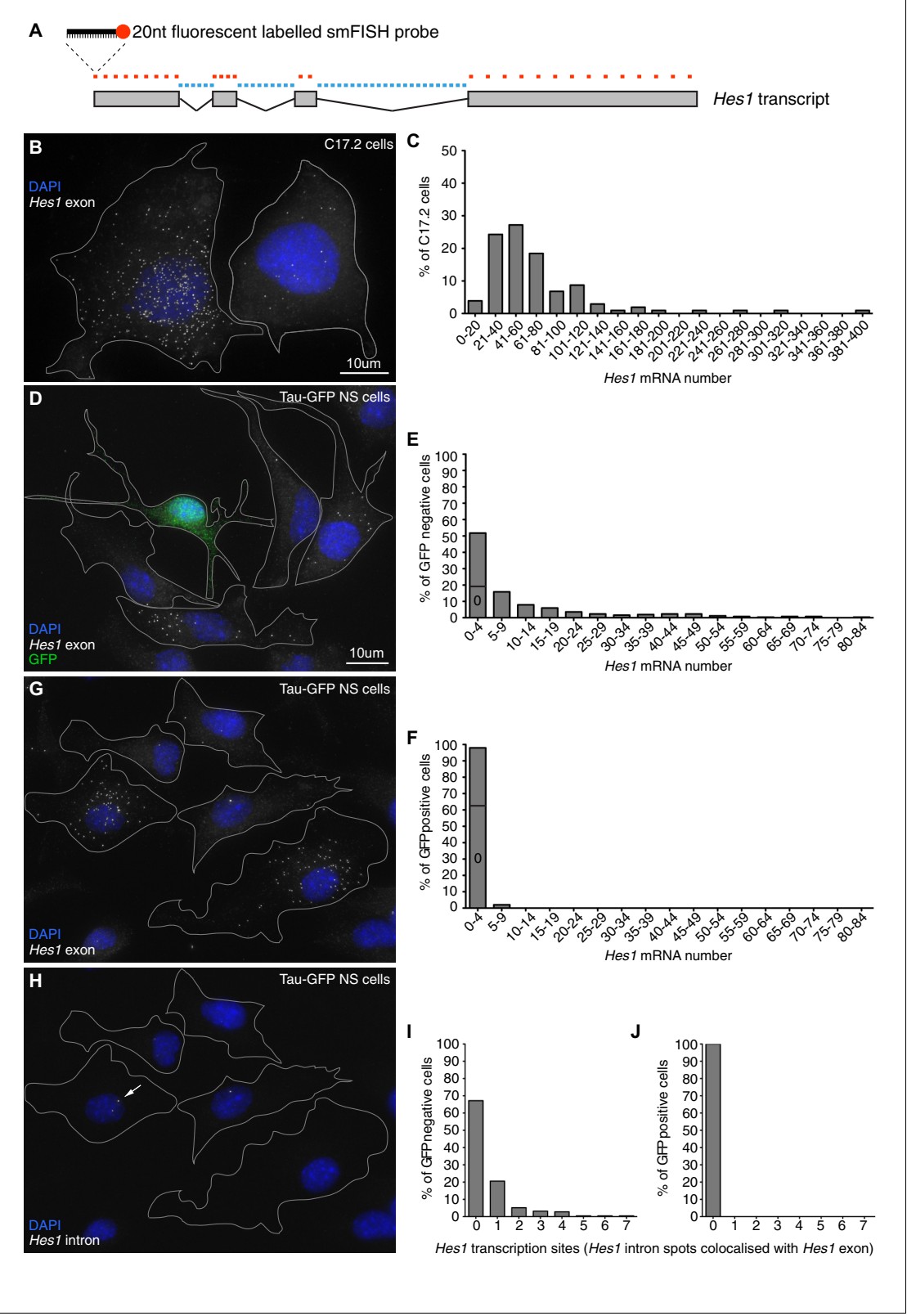

**Figure 1.** Absolute quantification of mRNA copy number and active transcription sites. (**A**) Schematic showing the mouse *Hes1* primary transcript, with 29 smFISH probes targeting exonic sequence (red) and 38 smFISH probes targeting intronic sequence (blue). (**B**) smFISH for *Hes1* mRNA (white) in C17.2 cells; nuclei are stained with DAPI (blue). (**C**) Quantification of *Hes1* mRNA number in C17.2 cells (n=103). (**D**) smFISH for *Hes1* mRNA (white) in Tau-GFP NS cells; nuclei are stained with DAPI (blue), and GFP protein is stained with rabbit anti-GFP > anti-rabbit Alexa Fluor 488 (green). The field
*Figure 1 continued on next page*

*Figure 1 continued*

contains one GFP positive cell, as determined by GFP immunofluorescence intensity analysis (*Figure 1—figure supplement 3*). (E) Quantification of *Hes1* mRNA number in GFP negative NS cells (n=253). (F) Quantification of *Hes1* mRNA number in GFP positive neuronal differentiated NS cells (n=49). (G, H) Double smFISH for *Hes1* mRNA (G) and *Hes1* intron (H) in the same Tau-GFP NS cells; arrow indicates an active transcription site, marked by colocalised exonic and intronic *Hes* smFISH signals in the nucleus. (I, J) Quantification of *Hes1* active transcription sites in GFP negative NS cells (I, n=253) and in GFP positive neuronal differentiated NS cells (J, n=49). Source data contained in *Figure 1—source data 1*.

The following source data and figure supplements are available for figure 1:

**Source data 1.** mRNA counts by smFISH in C17.2 and NS cells.
**Source data 2.** Quantification of miR-9 sensor activity.
**Source data 3.** Validation of smFISH accuracy.
**Source data 4.** Quantification of Tau-GFP signal intensity.
**Source data 5.** Quantification of miR-9 copy number in C17.2 cells by qRT-PCR.
**Figure supplement 1.** Quantification of miR-9 sensor activity in neural progenitor cells versus Tuj1 positive neurons.
**Figure supplement 2.** Validation of smFISH accuracy.
**Figure supplement 3.** Quantification of GFP signal intensity in Tau-GFP NS cells.
**Figure supplement 4.** Method to quantify miR-9 copy number.
**Figure supplement 5.** Quantification of miR-9 copy number.

mostly negative for *Hes1* mRNA transcript, indicating that the gene is switched off upon differentiation (average 0.8 copies per cell) (*Figure 1D and F*). This was confirmed by using an intronic probe for *Hes1* which showed transcription in GFP negative cells (*Figure 1H*, corresponding exonic shown in G, I) but no transcription in GFP positive neuronally differentiated cells (*Figure 1J*).

For mature miR-9, we used a stem-loop Taqman qRT-PCR and synthetic miR-9 (*Figure 1—figure supplement 4*) to estimate the absolute copy number of endogenous miR-9, expressed as a population average copy number per cell in C17.2 cells. Mature miR-9 was present in an average copy number of 1100 ± 20 (S.E.M) molecules per cell, significantly lower than other reported microRNAs and surprisingly low considering the hundreds of miR-9 targets (*Bonev et al., 2011*) (*Figure 1—figure supplement 5*).

Finally, to quantify the abundance of endogenous HES1 protein per nucleus in NS cells we generated two stable HES1 reporter cell-lines: VENUS:HES1 driven either by the constitutive *UbC* promoter or the 2.7 kb *Hes1* promoter (*Figure 2A,B*). FCS was used to count the number of VENUS:HES1 fluorophores (see Experimental methods). This gave an estimate of 3660 ± 120 (S.E.M) molecules of VENUS:HES1 per nucleus when driven by the *UbC* promoter (*Figure 2D* and *Figure 2—figure supplement 1*). We next used western blotting to find the ratio of expression between VENUS:HES1 and endogenous HES1 (*Figure 2C*) and subsequently normalised to the percentage of VENUS:HES1 positive cells in the population. This gave variable ratios between experiments and showed an approximate ratio of 1:0.8 ectopic to endogenous HES1. This suggests an average abundance of endogenous HES1 of 2930 molecules per nucleus which is significantly lower than the median protein abundance of 50,000 copies per cell previously reported from global quantification studies (*Schwanhausser et al., 2011*). Furthermore, FCS quantification of VENUS:HES1 driven by the *Hes1* promoter gave a similar value, that is 2450 ± 190 (S.E.M) molecules per nucleus (*Figure 2D*). These complementary approaches highlight the low average abundance and high variability of HES1 molecules in the cell.

We then investigated whether a stochastic or deterministic simulation of the *Hes1*/miR-9 interaction model is a better fit to the experimental data of HES1 protein and mRNA dynamics at the

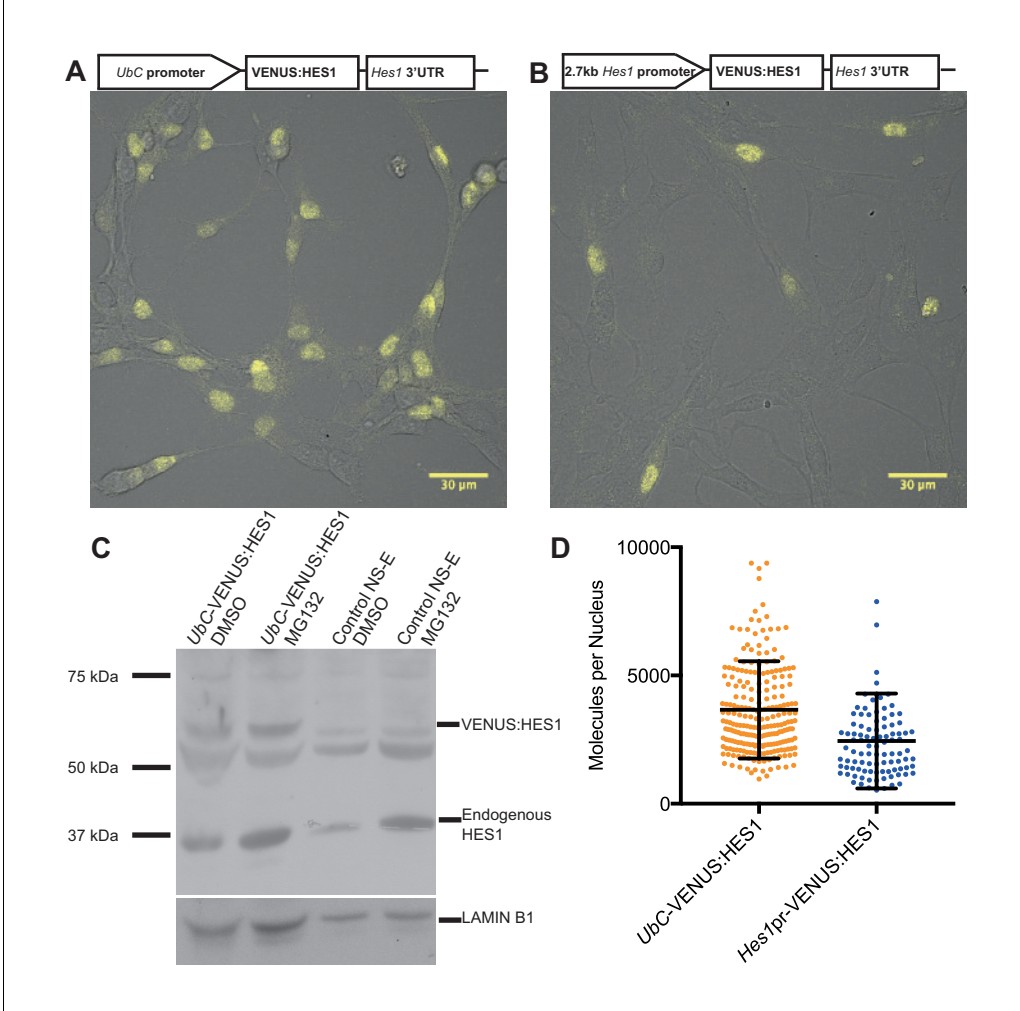

**Figure 2.** Quantifying HES1 protein number with FCS. Schematic and image of stably infected NS-E cells containing (**A**) *UbC*-VENUS:HES1 reporter construct and (**B**) 2.7 kb-*Hes1pr*-VENUS:HES1 reporter construct. VENUS:HES1 signal in yellow. Scale bars show 30 μm. (**C**) HES1 western blot with LAMIN B1 loading control. Nuclear lysates of NS-E cells stably infected with *UbC*-VENUS:HES1 or control NS-E cells. Both cell-lines treated with DMSO or 10 μM MG132 proteasome inhibitor for 3 hr. MG132 treated cells show increased HES1 and VENUS:HES1. (**D**) FCS quantification of VENUS:HES1 molecules per nucleus in stable *UbC*-VENUS:HES1 and 2.7 kb-*Hes1pr*-VENUS:HES1 reporter NS-E cell-lines. 236 cells (mean 3660 ± 120 (S.E.M) molecules) and 98 cells (mean 2450 ± 190 (S.E.M) molecules) were analysed in 4 and 2 experiments respectively. Error bars show S.D. FCS data and Western blots at different exposures are contained in *Figure 2—source data 1*.

The following source data and figure supplement are available for figure 2:

**Source data 1.** FCS data and Western blots.

**Figure supplement 1.** Fluorescence Correlation Spectroscopy technique.

single-cell and population level. We simulated the model of the *Hes1*/miR-9 network using the default parameters of Goodfellow et al. (*Goodfellow et al., 2014*) (parameter set 1 – *Appendix 1—table 1*) in both a deterministic and stochastic framework (using the dSSA algorithm). While oscillations in the deterministic model remain coherent (*Figure 3B*), the stochastic model shows variability in its dynamics and period (*Figure 3C*). Single-cell luciferase imaging of primary NS cells containing a Luciferase2-HES1 (LUC2:HES1) fusion BAC reporter (*Imayoshi et al., 2013*) shows a high degree of noise evident as peak-to-peak variability in amplitude and period. This was also evident in our

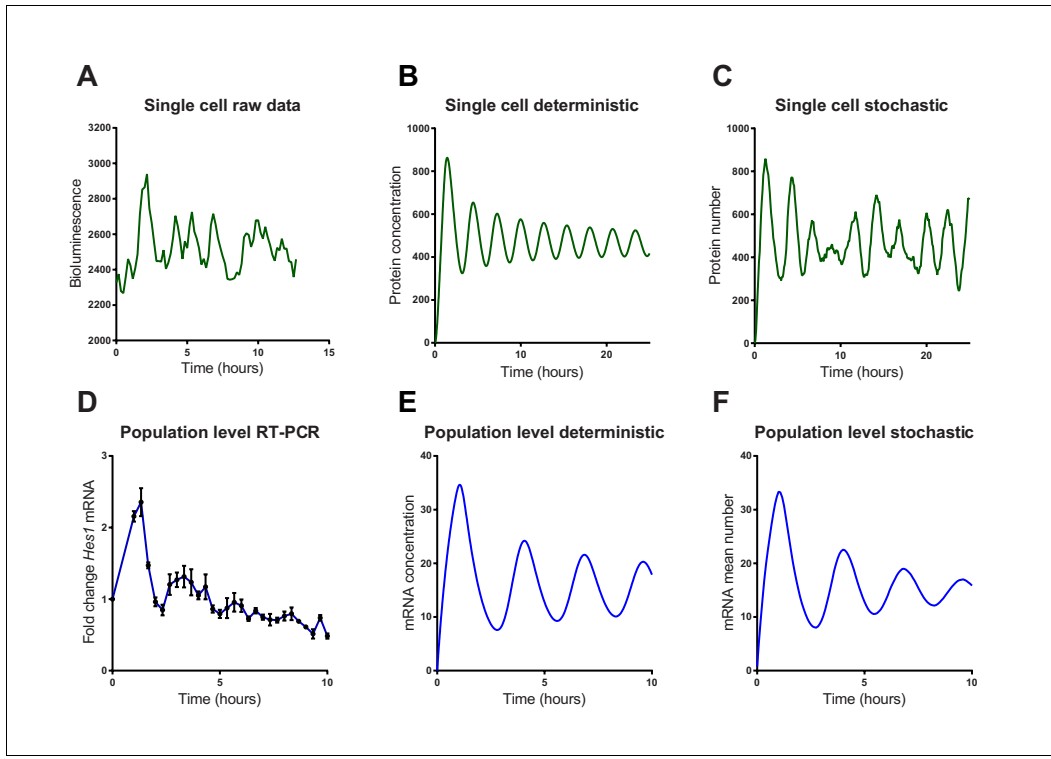

**Figure 3.** Stochastic simulations of HES1 oscillations match the experimental data better. (**A**) Time series of LUC2: HES1 reporter protein expression in a single primary NS cell determined by bioluminescence imaging. (**B**) Deterministic simulation of *Hes1*/miR-9 network at a single-cell level, plot shows HES1 protein concentration as a function of time. (**C**) Stochastic simulation of *Hes1*/miR-9 network with the dSSA at a single cell level. (**D**) q-PCR quantification of endogenous *Hes1* mRNA in C17.2 cells at the population level following serum synchronization. Error bars show S.E.M of 2 technical replicates of 2 biological experiments. (**E**) Population level deterministic simulation of *Hes1*/miR-9 network through averaging 2000 single-cells, with *Hes1* mRNA shown as a function of time. (**F**) Population level stochastic simulation of Hes1/miR-9 network with the dSSA and averaging over 2000 cells, with *Hes1* mRNA shown as a function of time. All simulations performed with parameter set 1 (***Appendix 1—table 1***) with system size $\Omega = 1$ and initial conditions m(0) = p(0) = r(0) = 0. Note that the computational simulations are less damped than the experimental data, presumably because other sources of variability in experimental data lead to greater damping. MATLAB code and experimental data contained in ***Figure 3—source data1***.

The following source data and figure supplements are available for figure 3:

**Source data 1.** MATLAB code to simulate the Hes1/miR-9 network with a deterministic or stochastic model (with dSSA).

**Source data 2.** Multiple time series examples of single-cell experimental data.

**Figure supplement 1.** Bioluminescence images from a single primary LUC2:HES1 NS cell cultured in proliferative conditions.

**Figure supplement 2.** Multiple single-cell time series of LUC2:HES1 reporter protein expression in primary NS cells determined by bioluminescence imaging.

experiments (***Figure 3A*** and ***Figure 3—figure supplement 1*** and ***2*** for images and traces of multiple cells). At the population level, analysed by qRT-PCR of serum synchronised C17.2 cells, oscillations in *Hes1* mRNA quickly dampen, presumably because cells drift out of synchrony (***Figure 3D***). The average expression of a population of 2000 stochastic simulations using the dSSA shows faster

dampening (*Figure 3F*) than deterministic simulations (*Figure 3E*). Thus, the stochastic simulation matches the experimental data better, both at the single-cell and the population level.

## Stochasticity spreads the timing of differentiation

By measuring the time taken to reach a low HES1 state in multiple simulations we can describe the distribution of timing events expected in a population of cells. The contribution of stochasticity was investigated by varying the system size parameter $\Omega$, which changes the number of molecules in the system. All analysis was performed using parameters for which the amplitude of oscillations is large and the transition to the low HES1 state is very clear, (parameter set 2 – *Appendix 1—table 1*). Once the HES1 protein expression reaches a low state we observed that it never again exceeds a low threshold of $20x\Omega$. We therefore defined 'differentiation' as the time taken for HES1 protein to decrease below this threshold.

In deterministic simulations, all cells that start with the same parameters and conditions differentiate at the same time. At a high system size ($\Omega=50$), corresponding to high molecule number (mRNA~500; protein and miR-9~30,000), there is low variability in the stochastic dynamics and oscillations appear regular (*Figure 4A*). The time taken to reach the low HES1 protein threshold was calculated for 2000 simulations using the dSSA, and the distribution of times to reach the low HES1 state is narrow for $\Omega=50$ (*Figure 4B*), approaching the deterministic solution.

As the system size is decreased to $\Omega=5$ (mRNA~50; protein and miR-9~3000) and then further to $\Omega=1$ (mRNA~10; protein and miR-9~600) molecular fluctuations cause increasingly varying amplitude and period of HES1 oscillations within each simulation (*Figures 4D and G*, respectively). At the same time, the distribution in the timing of differentiation of 2000 independent simulations becomes increasingly broad as the system size decreases (*Figures 4E and H*). Similar results were obtained with the CLE (*Figures 4C-I*). We conclude that the greater random molecular fluctuations caused by decreased molecular copy number acts to increase the uncertainty in the timing to reach a low HES1 protein state in each cell, such that differentiation events become distributed in the population over a longer period of time.

## Stochasticity shifts the average onset of differentiation

When we reduced the system size we unexpectedly found that the average of the distribution also became gradually shifted towards earlier times (e.g. compare *Figure 4B* with *Figure 4E and H*, where scaling of mean with system size shown in *Figure 4—figure supplement 1*). This shift indicates that stochasticity also causes a systematic change in timing, whereby the low HES1 state is reached on average earlier at low system sizes, even though the parameters of the model remain exactly the same. The origin of the systematic change in timing can be understood by considering how the production of miR-9 changes in the presence of stochasticity.

Transcription of miR-9 is repressed by HES1 protein, and consequently the production of miR-9 (*r*) increases when HES1 protein levels are low, assuming activation in the absence of repression. The production of miR-9 is therefore intrinsically linked to HES1 protein, and stochastic variability in HES1 protein causes stochasticity in the production of miR-9 (*Figure 5A*, miR-9 production is blue line). As the repression of miR-9 transcription by HES1 protein is modelled as a nonlinear Hill-type function, the higher incidence of low HES1 troughs at lower system sizes gives rise to large spikes of miR-9 production (compare blue line in *Figure 5A* with *Figure 5B*). In turn, this leads to a faster increase of the stable mature miR-9 (red line in *Figure 5A and B*) and a reduction in the average time to differentiation.

However, the systematic change in timing caused by stochasticity is not guaranteed to speed up the system. Instead of low troughs in HES1 causing spikes in miR-9 production, in an alternative parameter set the peaks in protein could cause a disproportionate repression in miR-9 production. This occurs, for example, when the transcriptional repression threshold of miR-9 production is roughly twice that of the *Hes1* promoter ($p_1$ = 750), as shown in *Figure 5C*. In this scenario the systematic effect of stochasticity on timing is reversed and the low system size simulations reach the low HES1 state at a later time. Note that the relative effect on timing is smaller, but this may be expected as HES1 is only weakly repressing miR-9 transcription for this parameter setting.

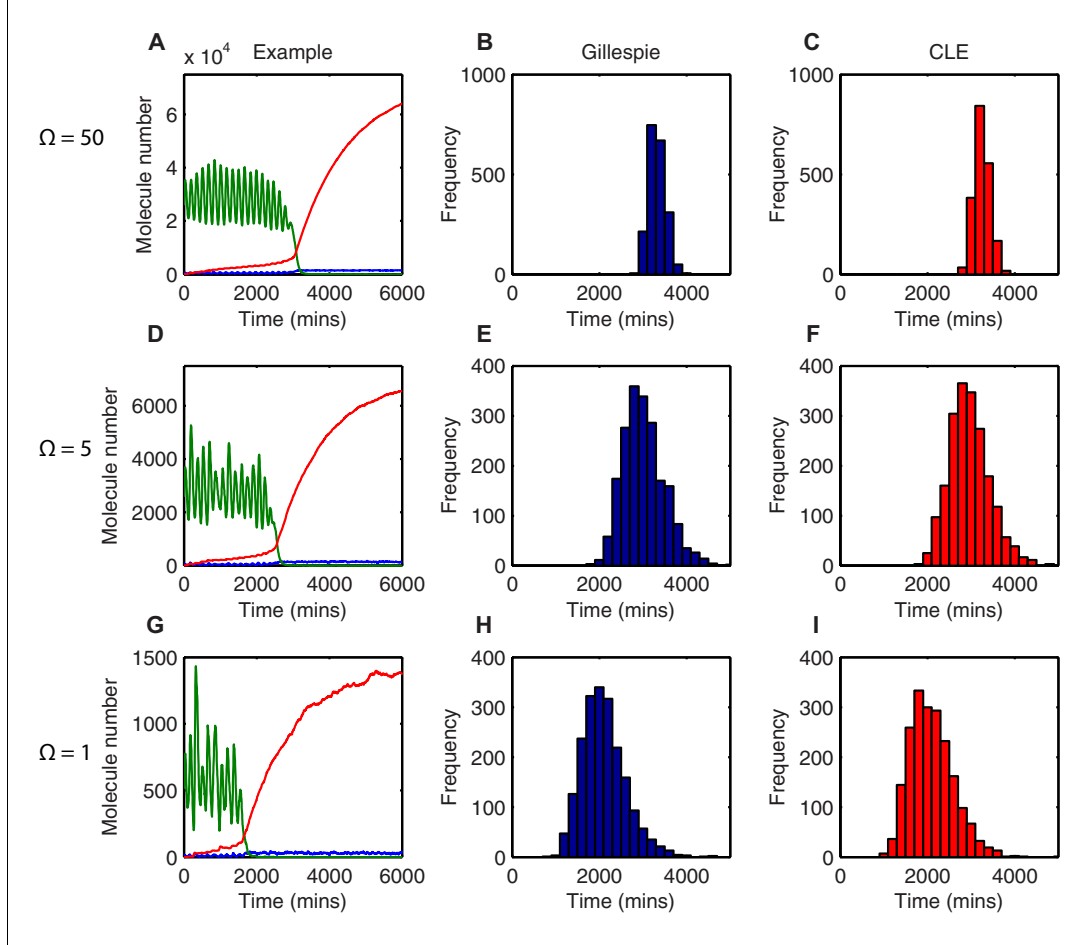

**Figure 4.** The distribution of time-to-differentiation widens at low system sizes. (A, D, G) Examples of stochastic simulations using the dSSA at Ω=50, 5 and 1, respectively. Blue, mRNA; green, protein; red, miR-9. (B, E, H) The distribution of times to reach the low HES1 protein state of 2000 simulations using the dSSA. (C, F, I) The distribution of times to reach the low HES1 protein state of 2000 simulations using the CLE. Note that even at low system size, the CLE is in excellent agreement with the Gillespie dSSA, which is consistent with previous studies showing that the CLE can be highly accurate when reactions only involve the creation or degradation of one species and act to either increase or decrease the molecule number by one (**Grima et al., 2011**) and confirms the results. All simulations performed with parameter set 2 (**Appendix 1—table 1**) and initial conditions m(0) = 20xΩ, p(0) = 400xΩ and r(0) = 0. MATLAB code contained in **Figure 4—source data 1**.

The following source data and figure supplement are available for figure 4:

**Source data 1.** MATLAB code for measuring time to differentiation and systematic shift in mean.

**Figure supplement 1.** The mean of the time-to-differentiate as a function of the system size.

## Increased miR-9 concentration shifts timing and reduces the spread of differentiation

The concentration of miR-9 is a key driver of the timing to differentiation of a single cell (single simulation) in the deterministic model (**Goodfellow et al., 2014**). We therefore tested the effect of altering the concentration of this key component in the stochastic model and in a population-level simulation.

In the stochastic model increasing the amount of miR-9 brought forward and reduced the spread of time to differentiation events in a simulated population (**Figure 6A–C**). To experimentally validate this theoretical finding we measured the timing and spread of differentiation in a population of C17.2 and murine Tau-GFP NS cells transfected with miR-9 mimic. Increasing miR-9 in C17.2 neural progenitor cells increased the number of Tuj1-positive neurons earlier than in control cultures after

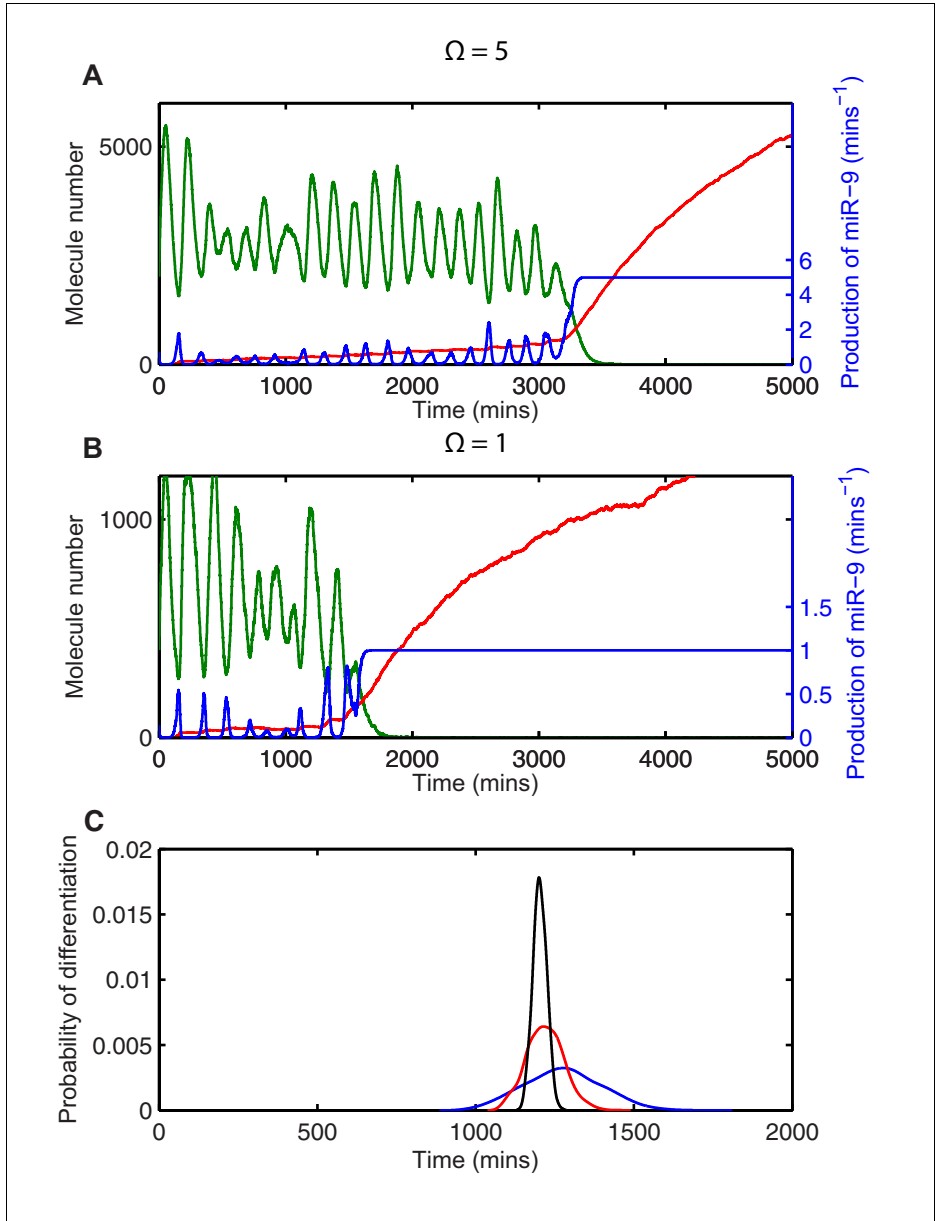

**Figure 5.** Stochasticity causes systematic changes to the mean time-to-differentiation. The time course of HES1 protein and miR-9 expression, compared with the production of miR-9 at a system size of (**A**) $\Omega=5$ or (**B**) $\Omega=1$ and simulated with the CLE. Green, protein; red, miR-9; blue, production of miR-9. All simulations performed with parameter set 2 (*Appendix 1—table 1*) and initial conditions m(0) = 20x$\Omega$, p(0) = 400x$\Omega$ and r(0) = 0. (**C**) The distribution of time to reach the low HES1 protein state when $p_1$ = 700, $\alpha_r$ = 0.25. Black, $\Omega=50$; red, $\Omega=5$, blue, $\Omega$ = 1. Note that here differentiation is defined to reach a miR-9 high state (r>170), as the transition to the low protein state is much less well defined. MATLAB code contained in *Figure 5—source data 1*.

The following source data is available for figure 5:

**Source data 1.** MATLAB code to simulate *Hes1*/miR-9 network using the CLE with tracked miR-9 production and noise-induced delayed differentiation.

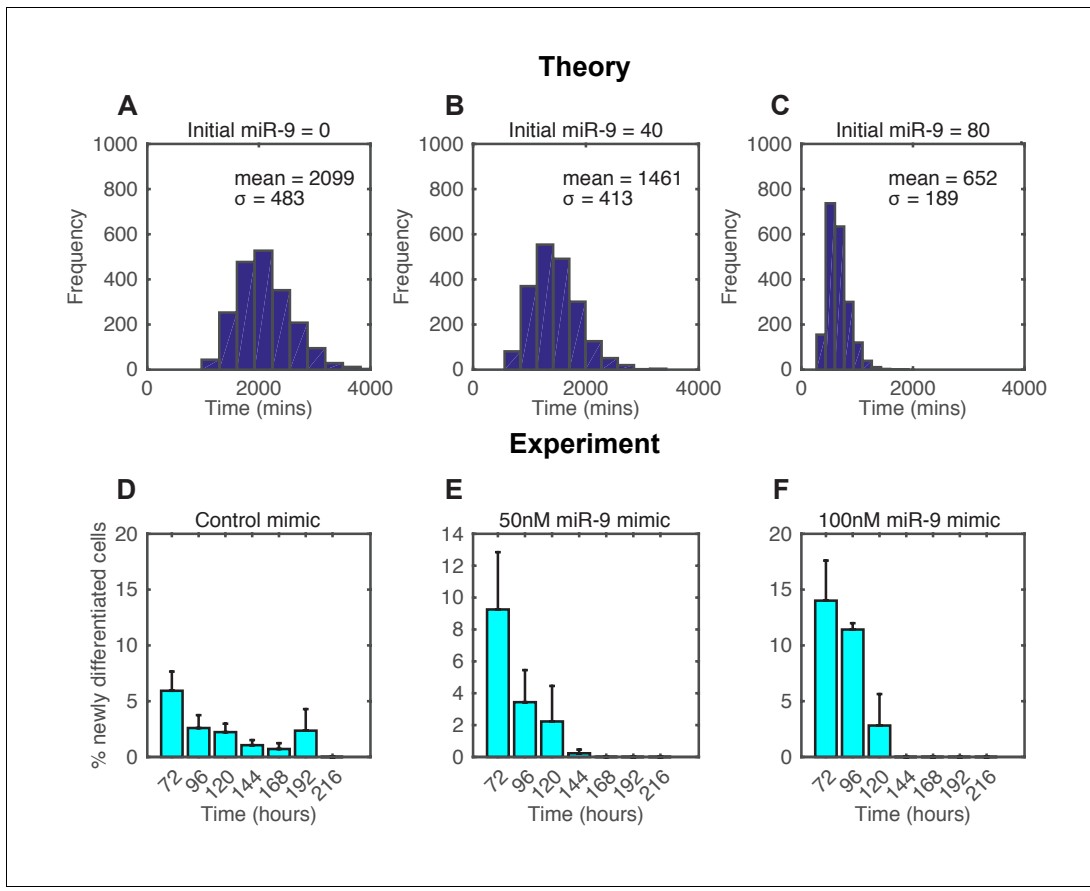

**Figure 6.** The spread of time-to-differentiation decreases in both the computational model and experiments with increased initial miR-9. (A, B, C) The distribution of times to reach the low HES1 state of 3000 simulations using initial conditions of miR-9 r(0) = 0, 40 and 80, respectively. The initial conditions of mRNA and protein were m(0) = 20xΩ, p(0) = 400xΩ. All simulations performed with parameter set 2 (*Appendix 1—table 1*) with system size Ω = 1 and using the CLE. (D, E, F) Newly differentiated Tuj1 positive C17.2 cells compared to previous time point as a percentage of the population. Cells were transfected with 100 nM control mimic or 50 nM and 100 nM miR-9 mimic, respectively. Cells were cultured in serum-free differentiation conditions for the time shown. 5 images analysed per condition in at least 2 experiments. MATLAB code and experimental data contained in *Figure 6— source data 1*.

The following source data and figure supplements are available for figure 6:

**Source data 1.** MATLAB code to simulate timing to differentiation with different initial conditions of miR-9.
**Source data 2.** Data for the speed up of differentiation in C17.2 cells with added miR-9.
**Source data 3.** Data for the speed up of differentiation in NS cells measured with FACS.
**Figure supplement 1.** Increased miR-9 shifts timing of differentiation earlier in C17.2 neural progenitor cells.
**Figure supplement 2.** Increased miR-9 shifts timing of differentiation earlier in NS cells.

several days, without affecting the total cell number (*Figure 6—figure supplement 1*). The shorter time to differentiate was confirmed by FACS analysis of Tau-GFP NS cells treated again with miR-9 mimic (*Figure 6—figure supplement 2*). Transfecting the C17.2 cells with a control mimic and measuring the number of Tuj1-positive neuronal cells showed a broad spread in the timing of differentiation (standard deviation = 48 hr, *Figure 6D*), as expected from a stochastic process. When cells were transfected with 50 nM or 100 nM of miR-9 mimic the average time to differentiation

decreased and the spread became narrower in a miR-9 initial concentration dependent way (standard deviation = 24 hr and 17 hr respectively, *Figure 6E and F*). Thus, both computational prediction and experimental validation suggest that the initial concentration of a single species, miR-9, is sufficient to change the variance in the time to differentiation.

## Stochasticity increases the robustness of the progenitor state

The progenitor state needs to be robust to avoid premature depletion of progenitors during the lengthy course of development. In order to investigate the robustness of the *Hes1*/miR-9 regulatory network we compared the termination of HES1 oscillations in stochastic and deterministic frameworks. This comparison was carried out systematically in the space of the parameters $r_1$ and $m_1$ using the linear noise approximation (LNA) (see Appendix for details). These control how much miR-9 is required to repress *Hes1* mRNA translation by half, and how nonlinear (sigmoidal) the translational repression is as a function of miR-9. We have previously shown that translational repression of *Hes1* mRNA by miR-9 is particularly important in terminating the *Hes1* oscillator with a low HES1 protein level (*Goodfellow et al., 2014*).

Different behaviours between the deterministic and the stochastic model are shown in this parameter space (*Figures 7A and B*). In the first region, where translational repression is highly nonlinear (high $m_1$) and translational repression strength is low (high amount of miR-9 required; high $r_1$), miR-9 never accumulates to the levels required to terminate oscillations (*Figure 7A* area 1). As a result, deterministic simulations in this parameter region show self-sustained limit cycle oscillations ($m_1$ = 3.5, $r_1$ = 900) (*Figure 7C*). The equivalent stochastic simulations using the dSSA also display sustained periodic behaviour (*Figure 7D*). This shows that when the translational repressive effect of miR-9 is weak, oscillations are not terminated in either deterministic or stochastic settings.

In the second region, (Area 2 in the deterministic model; *Figure 7A*), defined by highly nonlinear translational repression (high $m_1$) and high translational repression strength (low amount of miR-9 required to repress translation; low $r_1$), both deterministic and stochastic models produce HES1 protein levels that decrease and reach a steady state (*Figure 7E and F*, respectively). The power spectrum of the stochastic simulations at $m_1$ = 3.5, $r_1$ = 600 shows no peak, and hence the coherence (defined in Materials and methods, *Figure 7—figure supplement 1*) is zero (*Figure 7—figure supplement 2A*) The resulting behaviour of such a process is aperiodic (*Figure 7—figure supplement 2B*); therefore even though HES1 protein still shows some fluctuations (*Figure 7F*), the lack of periodicity allows us to conclude that oscillations are terminated in the stochastic setting. Thus, in this area oscillations are terminated in both models.

Of particular interest is the third region (Area 3 in the stochastic model; *Figure 7B*), which is characterised by less nonlinear repression (low $m_1$) and low translational repression (i.e. high amount of miR-9 to repress translation by half, high $r_1$). In this area of parameter space the periodic behaviour is distinct between the deterministic and stochastic framework. The deterministic solution in this regime exhibits damped protein oscillations that eventually reach a steady state (*Figure 7G*). In contrast, the time series of a stochastic simulation using the dSSA at this parameter configuration shows that even though miR-9 accumulates over time, it is unable to terminate protein oscillations (*Figure 7H*). When the system oscillates in the stochastic regime ($m_1$ = 1.5, $r_1$ = 600), there is a well-defined peak in the power spectrum (*Figure 7—figure supplement 2C*). The peak at 0.04 min$^{-1}$ corresponds to a period of 160 min in the time series (*Figure 7—figure supplement 2D*). Thus, this region represents an expansion of the range of parameters in which HES1 protein oscillates.

We propose that stochasticity has made the oscillatory state more robust to an increase in miR-9 in the sense that the stochastic HES1 oscillator continues to operate under conditions (i.e. parameter values) where the deterministic equivalent will have stopped. It is still possible for the accumulation of miR-9 to terminate HES1 oscillations in the stochastic model but the parameter space in which this happens is reduced.

## Robustness of timing to perturbations from asymmetric inheritance at cell division

Having seen that stochastic oscillations are inherently more robust in that they are observed for an increased parameter space, we also tested whether they would show increased robustness to external perturbations. Such perturbations can arise by the unequal distribution of molecules during cell

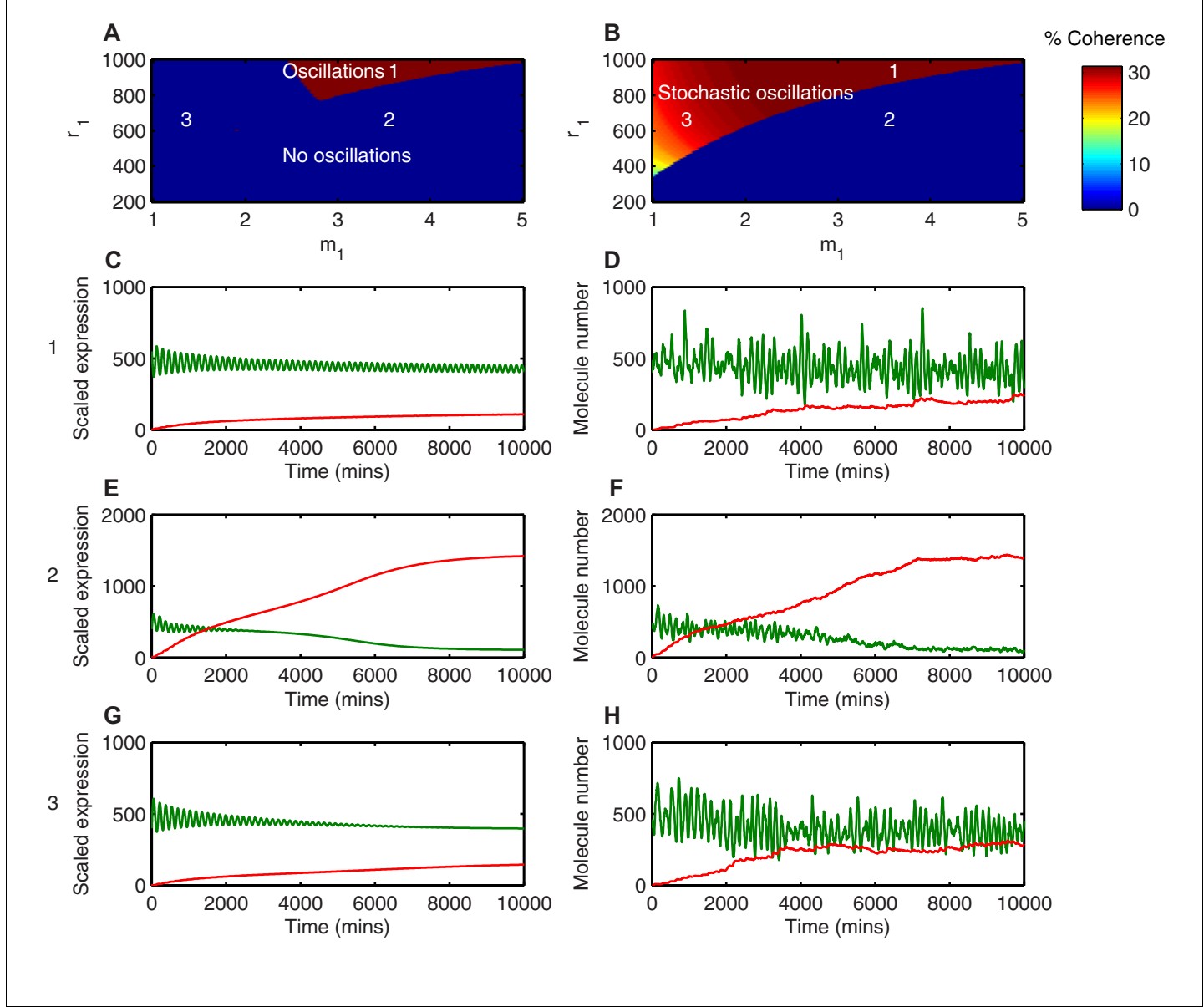

**Figure 7.** Stochasticity increases the parameter space where HES1 oscillates. (A) The areas of HES1 protein oscillations in the deterministic system, as a function of translation repression Hill co-efficient ($m_1$) and translation repression threshold ($r_1$). (B) The areas of HES1 protein oscillations in the stochastic system at the steady state as calculated by the LNA. The colour scale represents the coherence of the stochastic oscillations. (C, E, G) Deterministic simulations of the system for point 1 ($m_1$ = 3.5, $r_1$ = 900), point 2 ($m_1$ = 3.5, $r_1$ = 600) and point 3 ($m_1$ = 1.5, $r_1$ = 600), respectively. Green, Protein; red, miR-9. (D, F, H) Stochastic simulations (dSSA) of the system for point 1 ($m_1$ = 3.5, $r_1$ = 900), point 2 ($m_1$ = 3.5, $r_1$ = 600) and point 3 ($m_1$ = 1.5, $r_1$ = 600), respectively. All simulations performed with parameter set 1 (*Appendix 1—table 1*) and initial conditions m(0) = 20x$\Omega$, p(0) = 400x$\Omega$ and r(0) = 0. MATLAB code contained in *Figure 4—source data 1*.

The following source data and figure supplements are available for figure 7:

**Source data 1.** MATLAB code to scan parameters for deterministic and stochastic oscillations.
**Source data 2.** MATLAB code to generate power spectra of examples.
**Figure supplement 1.** Coherence of a power spectrum.
**Figure supplement 2.** Random fluctuations are distinct from stochastic oscillations.

division and can be modelled by changing the initial conditions of the simulations. We consider two models of cell division: one where molecular constituents are distributed equally between daughter cells (and hence initial conditions are the same at 50% of the parent cell) and another where molecules are randomly inherited.

First, we investigated how *Hes1* mRNA is actually partitioned in NS cells that are undergoing fate symmetric divisions (i.e. grown under proliferative divisions). Very few of these divisions partitioned *Hes1* mRNA equally and some showed a highly asymmetric distribution (*Figure 8A–D*). The majority of cell divisions (36/51) were mostly consistent with a binomial mode of division (p-value>0.05, binomial distribution with Bonferroni multiple testing adjusted p-value), with a few outliers. A binomial distribution would be expected if each mRNA molecule is partitioned independently into the two daughter cells and where the probability of inheriting an mRNA is equal to the ratio of the cell areas of the daughter cells. Previous studies in bacteria have also shown that partitioning is approximately binomial (*Golding et al., 2005*).

Using the binomial distribution we then calculated the impact of partitioning noise into the performance of the HES1 oscillator under deterministic and stochastic conditions. To generate initial

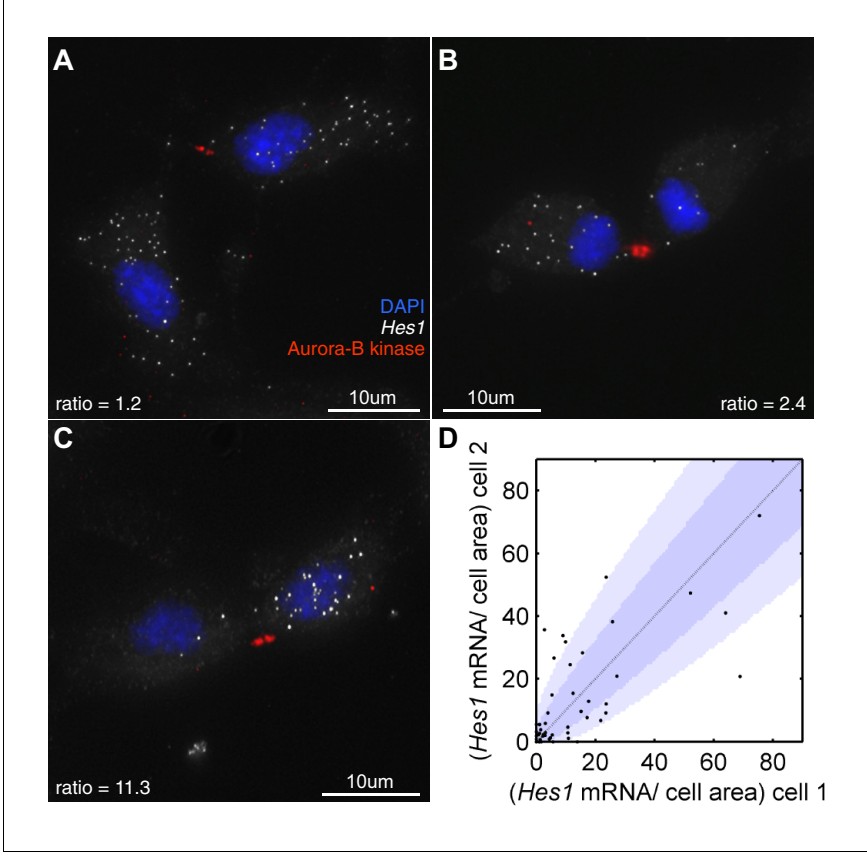

**Figure 8.** *Hes1* mRNA is inherited unequally at cell division. (**A**, **B**, **C**) smFISH with an exonic *Hes1* probe showing asymmetry in *Hes1* mRNA number (white) between sister NS cells, just prior to their complete separation at cytokinesis. Aurora-B kinase antibody staining (red) marks the midbody at cytokinesis; nuclei are DAPI stained (blue). (**D**) Quantification of *Hes1* mRNA to cell area ratios between sister cells from 51 NS cell divisions at cytokinesis, from one experiment. Dark blue area represents the 90% confidence bounds that the ratios are drawn from a binomial distribution, where the average amount of mRNA received by each cell is normalised to the relative cell areas. Light blue represents multiple testing Bonferroni corrected 90% confidence bounds that the ratios are drawn from a binomial distribution. mRNA count data contained in *Figure 8—source data 1*.

The following source data is available for figure 8:

**Source data 1.** smFISH mRNA counts of cells at cell division.

conditions we simulated a parent cell deterministically for 12 hr, starting with parameter set 2 and initial conditions mRNA = 20xΩ, protein = 400xΩ, miR-9 = 0. Using these parent cell conditions, we then generated daughter cells assuming either equal 50/50 inheritance (no heterogeneity) or a binomial distribution of inheritance for each species (schematic shown *Figure 9A*). *Figure 9* shows the

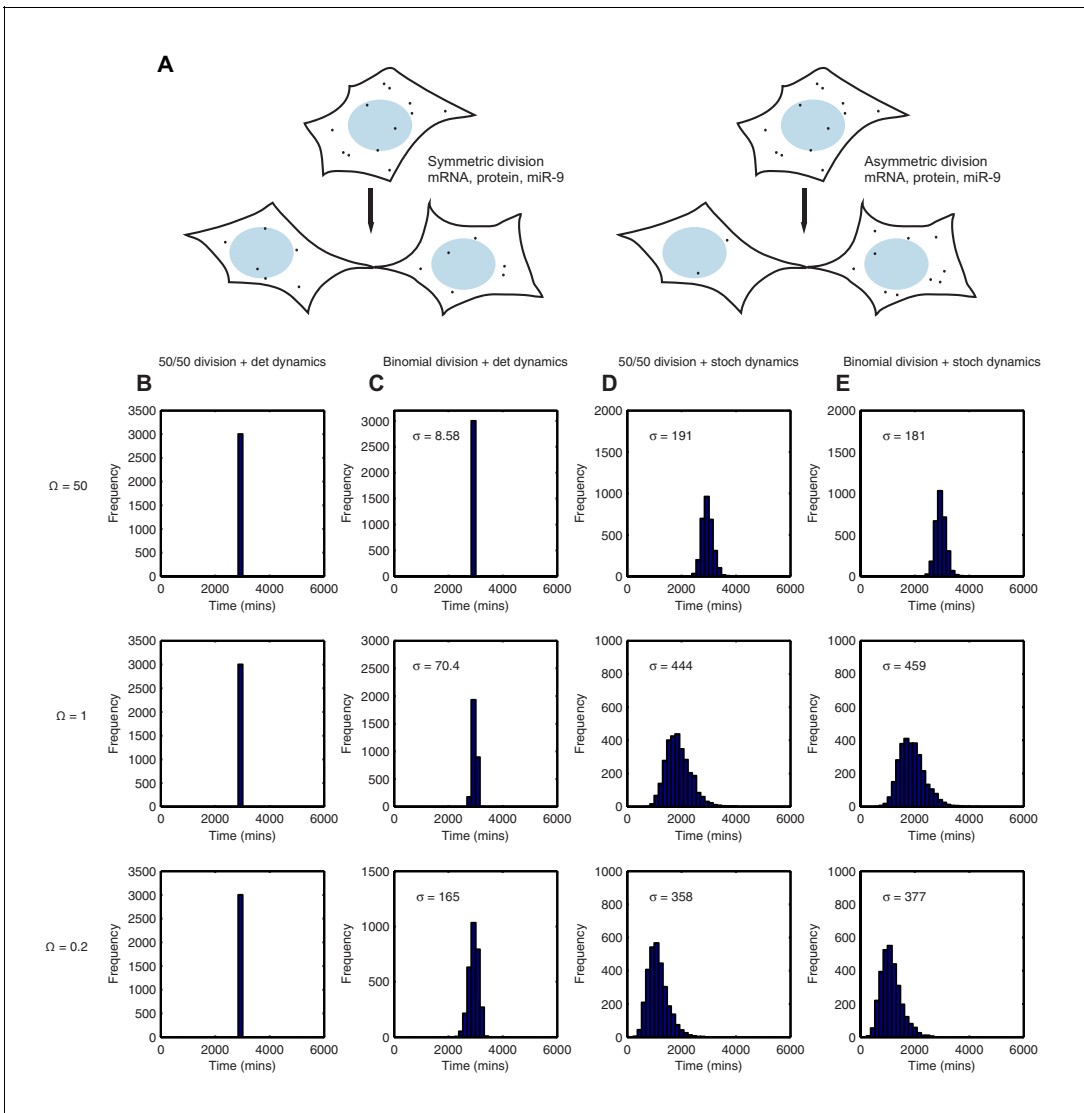

**Figure 9.** Stochastic system is robust to the binomial distribution of constituents at cell division. (**A**) Schematic to show the difference between symmetric and asymmetric division. (**B**) The distribution of time to reach the low HES1 state with 50/50 initial conditions using deterministic dynamics as system size of Ω = 50, 1 and 0.2, respectively. (**C**) The distribution of times to reach the low HES1 state of 3000 simulations with binomial initial conditions using deterministic dynamics at system size of Ω = 50, 1 and 0.2, respectively. (**D**) The distribution of times to reach the low HES1 state of 3000 simulations with 50/50 initial conditions using the CLE at system size of Ω = 50, 1 and 0.2, respectively. (**E**) The distribution of times to reach the low HES1 state of 3000 simulations with binomial initial conditions using the CLE at system size of Ω = 50, 1 and 0.2, respectively. All simulations performed with parameter set 2 (*Appendix 1—table 1*) and initial conditions m(0) = 20xΩ, p(0) = 400xΩ and r(0) = 0. The standard deviation of the distribution is denoted with σ. In the simulations shown, parent cells were simulated deterministically for 12 hr prior to 'division' however, the distributions remained statistically the same (p-value>0.05) when either 11 or 13 hr deterministic simulations prior to division were used (data not shown). MATLAB code contained in *Figure 9—source data 1*.

The following source data is available for figure 9:

**Source data 1.** MATLAB code for deterministic and stochastic dynamics with wither 50/50 or binomial division.

difference in the time distribution of differentiation assuming equal or binomial partitioning, with deterministic (*Figure 9B,C*) or stochastic (*Figure 9D,E*) dynamics.

With 50/50 inheritance at cell division (i.e. no heterogeneity) and deterministic dynamics there is no randomness in the system and hence all cells 'differentiate' at the same time (*Figure 9B*). The time is the same for all system sizes, as the number of molecules has no effect on the dynamics. Next, we introduce heterogeneity in the starting conditions of the cell, occurring only through random partitioning and using a binomial distribution. At a high system size, the distribution of time-to-differentiation is very narrow (standard deviation = 8.58) (*Figure 9C*). This makes sense, because the relative variance of a binomial distribution is lower at high copy number and hence at large molecule number the daughter cells have similar inheritance and therefore similar timing to differentiation. As the system size is lowered ($\Omega$=1 and 0.2), the variance increases and the variability in the daughter cell initial conditions causes the time-to-differentiation to become more spread out, compared either to the high system size (either mode of partitioning) or the low system size with equal partitioning (standard deviation = 165, p-value<0.05 for Kolmogorov-Smirnov test between $\Omega$ = 0.2 and 50) (*Figure 9C*). This finding denotes a lack of robustness to partitioning errors when one looks at the outcome of the HES1 oscillator at the population level.

We now consider the effect of unequal inheritance in the case of stochastic dynamics. At both high and low system size (low and high intrinsic noise, respectively) there is little difference between the distribution created with 50/50 (constant) or unequal (binomial) initial conditions as indicated by a low value of the Kolmogorov-Smirnov distance between the two distributions and a p-value indicating a non-significant difference (*Figure 9D and E*).

We conclude that the stochastic system is more robust than the deterministic system to noise introduced at cell division because it produces the same distribution of differentiation at the population level irrespective of whether the cells distribute molecules randomly or equally at cell division, at all system sizes. In other words, the stochastic system shows tolerance to the unequal distribution of species at division and maintains its performance. By contrast, in the deterministic framework the distribution of differentiation events is changed significantly by random partitioning of molecules, and this change increases as the system size is lowered.

## Discussion

At the molecular level, gene expression involves discrete reaction steps that result in the birth and death of individual molecules and it is therefore subject to the intrinsic noise generated by the random encounters of a finite number of interacting molecules (*Lestas et al., 2010*). It is widely assumed that such noise is detrimental to the robust performance of biological systems and indeed, a number of mechanisms to filter noise have been described (*Arias and Hayward, 2006*; *Bahar Halpern et al., 2015*). It remains an open question whether biological systems exploit noise, as is the question of whether noise optimization itself is an evolvable trait in development (reviewed in *Rao et al., 2002*; *Kaern et al., 2005*; *Balázsi et al., 2011*). Here, we have considered the influence of intrinsic noise generated by a low molecular number of interacting species, on the output of HES1 protein oscillations that are generated by delayed negative feedback and are tuned by miR-9. Our findings led us to propose that stochasticity generated by intrinsic noise offers a number of advantages to the differentiation process.

Modelling the differentiation process as an exit from the oscillatory state with low HES1 protein (*Goodfellow et al., 2014*), we found that in the stochastic framework the time taken to exit HES1 oscillations varied from cell to cell, even though they started from identical conditions. Hence, there was a distribution in the timing of differentiation. This is an important feature of the stochastic model, as it predicts that it is possible for an initially equivalent population of cells (e.g. clonally related, with the same transcriptional profile and under identical conditions) to change cellular identity at different times as a result of internal noise alone and not as a consequence of any other external influences; at the population level this would create a distribution in the timing of differentiation (*Gomes et al., 2011*; *Boije et al., 2014*). We suggest that it may be advantageous for differentiation events to have a broad distribution, as this could be used to generate cell-type diversity, if coupled with mechanisms of cell-type specification (*Qian et al., 2000*; *Temple, 2001*). However, a distribution in timing of differentiation is not only linked to the generation of different cell types, because when one considers the generation of a single particular cell type, there is again a distribution in the

timing of birth of this cell type (*Gaspard et al., 2008*; *Gomes et al., 2011*). This suggests that a distribution in the timing of differentiation is a widespread property, which is perhaps advantageous in allowing a time window for feedback control such that the right cell numbers are produced.

Our modelling also showed that there is a continuum of behaviours since the spread of the timing distribution is a function of the system size, where high system size gives rise to a narrow distribution and low system size to a broad distribution. We speculate that this may partly explain the existence of differentiation models that are highly deterministic, as for example in Drosophila neuroblasts which generate cells with a rigidly programmed sequence of fates (*Pearson and Doe, 2004*; *Kohwi and Doe, 2013*; *Rué and Martinez Arias, 2015*) and others that are highly stochastic, as the above mentioned example of vertebrate retinal development.

Our findings suggest that the incorporation of stochasticity into the HES1 oscillator model is advantageous because it increases the robustness of the progenitor state to intrinsic and extrinsic noise. The robustness to increased intrinsic noise is demonstrated by the increase in the region of the parameter space within which HES1 oscillates. This happens because in the parameter space near the oscillatory regime (Hopf bifurcation), the deterministic solution would exhibit damped oscillations but noise keeps pushing the system away from the steady state. This is consistent with previous reports that described noise induced oscillations (*McKane and Newman, 2005*; *Barrio et al., 2006*; *Galla, 2009*). However, while previous studies have looked at whether noise can induce oscillations in systems at steady state (statistically stationary), we have used LNA to look at whether initial transient oscillations are eventually terminated. In this way, we showed that miR-9 accumulation can terminate stochastic oscillations if the threshold of miR-9 required for translational repression is sufficiently low, and the Hill coefficient of repression is sufficiently high. This is important, as normal neural development relies on the balance of prolonged progenitor maintenance (i.e. robust HES1 oscillations) and differentiation (i.e. the ability to terminate these).

The outcome of deterministic dynamics is very different in symmetric versus randomly asymmetric divisions, meaning that deterministic dynamics are not robust to heterogeneities introduced at cell division. In particular, the narrow spread of differentiation that is produced by a deterministic molecular network would be impossible to maintain if the distribution of molecules at division is random and the system size low, both of which are experimentally observed. By contrast, the stochastic system has a more robust performance to extrinsic noise introduced by cell-division asymmetries, as the outcome does not change between equal or random division. Interestingly, at low system size the random partitioning of molecules at division produces a distributed time of differentiation, even when oscillations are modelled in a deterministic framework. We interpret this to mean that noise due to cell-division asymmetries has a similar effect to noise due to low system size on the outcome of the HES1 oscillator. This fits well with previous data on stochasticity introduced by partitioning errors (*Huh and Paulsson, 2011*). Our findings agree that noise introduced in this way is very difficult to distinguish from other sources of noise (*Huh and Paulsson, 2011*) and both could contribute in the heterogeneity of timing of clonal cells, although the effect does not appear to be additive.

In addition to the spread of differentiation events, the mean time-to-differentiation was also systematically affected in the stochastic framework, which can be explained by the dynamics of HES1 and miR-9 production in conjunction with the parameterisation of the system. This shows that in addition to parameters and initial conditions (*Goodfellow et al., 2014*), the average time-to-differentiation can also be tuned by the absolute number of molecules reacting in the system through the system size. The mean time-to-differentiation is important as it controls the total cellular number of a developed organ, along with other quantities such as the proliferation rate.

Here, we have assumed that the intrinsic noise is a result of low interacting numbers of molecules in the *Hes1*/miR-9 network. Indeed, this was experimentally supported by the absolute quantitation of *Hes1* mRNA, protein and miR-9. The average but absolute *Hes1* mRNA, protein and miR-9 numbers are consistent with a medium to low system size between $\Omega=5$ and $\Omega=1$, with variability observed both between cells and between different cell lines. In particular, while the *Hes1* mRNA copy number is close to the reported global median of 17 copies per cell, the HES1 protein abundance is well below the median value of 50,000 copies per cell (*Schwanhäusser et al., 2011*). The average measured miR-9 copy number (1100 copies) in C17.2 cells is also on the low side of the previously reported range of 10 to more than 30,000 copies per cell (*Chen et al., 2005*). We consider the abundance of HES1 protein and miR-9 to be low not only compared to average global abundancies, but also considering the number of their predicted molecular targets (3,074 for HES1,

(*Margolin et al., 2009*); and more than 500 for miR-9, (*Bonev et al., 2011*)). However, the low molecular copy number is not an exclusive source of cell-to-cell heterogeneity. The *Hes1* gene is also involved in the Notch signalling pathway of cell-cell communication, and other studies have considered the effect of noise in the Notch pathway (*Jenkins et al., 2015*). Additionally, spatial models of Notch signalling have considered the effect of stochastic motions of the nucleus within a gradient of Notch signalling molecules (*Aggarwal et al., 2016*). Such sources of heterogeneity are not mutually exclusive, and future studies may extend the model to include these effects.

In our model there is inherent stochasticity in every reaction controlling the birth and death of each species but the transcription rate can take a continuous range of values. Other studies of the *Hes1/Her1* network have considered binary promoter noise, where the transcription rate switches discontinuously between an ON and OFF state (*Lewis, 2003*; *Sturrock et al., 2013*; *Jenkins et al., 2015*). In the promoter ON/OFF model, periods of gene activity are interspersed by transcriptional silence, leading to bursty behaviour (*Suter et al., 2011*). Bursting models of transcription have been shown to fit the statistics of mRNA distributions measured by smFISH in multiple different systems (*Munsky et al., 2012*; *Neuert et al., 2013*; *Bahar Halpern et al., 2015*), and have also been fitted to live time series of transcription (*Harper et al., 2011*; *Corrigan and Chubb, 2014*; *Zechner et al., 2014*). Other recent studies have explicitly modelled the coupling of the *Hes1* system to the cell cycle (*Pfeuty, 2015a*, *2015b*), but carry the disadvantage that they contain many parameters that are not reliably known.

The challenge is to integrate these different phenomena, each potentially contributing a different source of noise, with the ultimate goal of connecting mechanistic explanations of developmental gene expression programmes with models of population dynamics. Our model provided an experimentally grounded and molecularly explicit theoretical framework to understand the underlying heterogeneity in the differentiation of clonal isogenic stem or progenitor cells, which have been uncovered by lineage tracing experiments in vivo, or in vitro cultures under the same conditions (*Rapaport et al., 2004*). Our work has also highlighted the need for integrated computational modelling and the value of an absolute quantitation in the experimental approach.

## Materials and methods

### Computational methods

Previous analysis focussed on a model of the *Hes1*/miR-9 network using delay differential equations (*Goodfellow et al., 2014*), which are themselves an extension of earlier models by *Monk et al (2003)*, as follows

$$\frac{dm}{dt} = G(p(t-\tau)) - S(r)m(t), \tag{2}$$

$$\frac{dp}{dt} = F(r)m(t) - \frac{\ln(2)}{\mu_p}p(t), \tag{3}$$

$$\frac{dr}{dt} = G_r(p(t)) - \frac{\ln(2)}{\mu_r}r(t), \tag{4}$$

where $m$, $p$ and $r$ are the concentrations of *Hes1* mRNA, HES1 protein and miR-9 respectively. The model parameters $\mu_p$ and $\mu_r$ are the half-lives of HES1 protein and miR-9 degradation, respectively. The quantities $G(p(t-\tau))$, $S(r(t))$, $F(r(t))$ and $G_r(p(t))$ are Hill-type functions that control the rates of synthesis and degradation. The variable $\tau$ indicates the time delay in the repression of *Hes1* transcription by HES1 protein, accounting for the total time taken from mRNA transcriptional initiation to HES1 protein returning to the nucleus to repress transcription. For a given set of initial conditions and parameters the solution of these differential equations is always the same, and they are therefore described as deterministic.

HES1 protein represses *Hes1* mRNA production through binding to the promoter, and this is modelled with a monotonically decreasing Hill function. In the model it takes the form

$$G(p(t-\tau)) = \frac{\alpha_m}{1 + (p(t-\tau)/P_0)^{n_0}}, \tag{5}$$

where $\alpha_m$ is the basal transcription rate of *Hes1* mRNA in the absence of HES1 protein (previously set to 1; [*Goodfellow et al., 2014*]).

The Hill function is characterized by two parameters: the repression threshold $p_0$, which represents the amount of protein required to repress transcription to half of basal levels, and the Hill coefficient $n_0$, which controls how steep (sigmoidal) the repression is as a function of protein concentration.

The degradation of *Hes1* mRNA is dependent on the levels of miR-9 through the Hill function $S(r)$

$$S(r) = b_l + \frac{(b_u - b_l)}{1 + (r/r_0)^{m_0}}, \tag{6}$$

The function $S(r)$ represents an effective degradation rate of *Hes1* mRNA, and $b_l$ and $b_u$ impose lower and upper bounds for *Hes1* mRNA half-life, respectively, which are set using half-lives measured in Bonev et al. (*Bonev et al., 2012*).

The function $F(r)$ represents translation repression of HES1 protein by miR-9, and the functional form is also a Hill-type function, as previously assumed in Osella et al. (*Osella et al., 2011*).

$$F(r) = \frac{1}{1 + (r/r_1)^{m_1}}, \tag{7}$$

Finally, the HES1 protein represses miR-9 production through binding to the promoter, and this is again modelled with a Hill function

$$G_r(p) = \frac{1}{1 + (p/p_1)^{n_1}}, \tag{8}$$

All analyses and simulations were performed using MATLAB 7.12 (RRID:SCR_001622)(MathWorks, US), and code is provided as associated figure source files. For deterministic simulations of the model, we used the MATLAB function 'dde23' with options 'RelTol' = $10^{-5}$ and 'AbsTol' = $10^{-8}$. For history vectors we used a single value repeated over prior time points. Numerical bifurcation analysis was performed using DDE-BIFTOOL (*Engelborghs et al., 2002*).

## Developing a stochastic model for the *Hes1*/miR-9 oscillator

In order to investigate the effects of intrinsic noise on the dynamics of differentiation we reformulated the *Hes1*/miR-9 model in a stochastic framework, which describes the system in terms of its underlying microscopic reactions. The numbers of molecules are increased or decreased through six reactions, namely the production and degradation of mRNA, protein and microRNA as follows:

$$\emptyset \overset{G}{\Longrightarrow} M \qquad\qquad \text{mRNA transcription} \tag{9}$$

$$\emptyset \overset{G_r}{\longrightarrow} R \qquad\qquad \text{miR-9 transcription} \tag{10}$$

$$M \overset{F}{\longrightarrow} M + P \qquad\qquad \textit{Hes} 1 \text{ translation} \tag{11}$$

$$M \overset{S}{\longrightarrow} \emptyset \qquad\qquad \text{mRNA degradation} \tag{12}$$

$$P \overset{\ln(2)/\mu_p}{\longrightarrow} \emptyset \qquad\qquad \text{protein degradation} \tag{13}$$

$$R \overset{\ln(2)/\mu_r}{\longrightarrow} \emptyset \qquad\qquad \text{miR-9 degradation} \tag{14}$$

The transcription reaction involves a delay, indicated by a double arrow, while all other reactions are assumed to occur instantaneously with the rates of reaction indicated above the arrows. The stochastic dynamics of the *Hes1*/miR-9 reaction system were captured by the chemical master equation (CME) for the number $n_m$ of mRNA molecules, $n_p$ of protein molecules and $n_r$ of miR-9 molecules.

The CME describes how the probability of the copy number $\mathbf{n} = (n_m, n_p, n_r)$ of the different molecular species evolves over time.

The six reactions in the system result in a stochastic process described by the CME

$$\begin{aligned}
\frac{d}{dt}P(\mathbf{n};t) = \; & \Omega(\mathbb{E}_m - 1)[S(n_r/\Omega)(n_m/\Omega)P(\mathbf{n};t)] + \Omega\frac{\ln(2)}{\mu_p}(\mathbb{E}_p - 1)[(n_p/\Omega)P(\mathbf{n};t)] \\
& + \Omega(\mathbb{E}_p^{-1} - 1)[F(n_r/\Omega)(n_m/\Omega)P(\mathbf{n};t)] + \Omega\sum_{\mathbf{n}'}G(n_p'/\Omega)(\mathbb{E}_m^{-1} - 1)[P(\mathbf{n};t;\mathbf{n}',t-\tau)] \\
& + \Omega\frac{\ln(2)}{\mu_r}(\mathbb{E}_r - 1)[(n_r/\Omega)P(\mathbf{n};t)] + \Omega(\mathbb{E}_r^{-1} - 1)[G_r(n_p/\Omega)P(\mathbf{n};t)]
\end{aligned} \tag{15}$$

where $P(\mathbf{n};t)$ is the probability of finding the system in state $\mathbf{n}$ at time $t$, and $P(\mathbf{n};t;\mathbf{n}',t')$ is the probability for the system to be in state $\mathbf{n}$ at $t$ and in state $\mathbf{n}'$ at time $t'$. $\mathbb{E}_m$, $\mathbb{E}_p$ and $\mathbb{E}_r$ are raising operators acting on functions of $n_m$, $n_p$ and $n_r$ via $\mathbb{E}_m g(n_m, n_p, n_r) = g(n_m + 1, n_p, n_r)$, $\mathbb{E}_p g(n_m, n_p, n_r) = g(n_m, n_p + 1, n_r)$ and $\mathbb{E}_r g(n_m, n_p, n_r) = g(n_m, n_p, n_r + 1)$. $\mathbb{E}^{-1}$ stands for the inverse operation. In the case of two-time quantities, e.g. $P(\mathbf{n};t;\mathbf{n}',t')$ the raising applies with respect to the second argument $n'$.

We have also developed a set of Chemical Langevin Equations (CLE) (**Brett and Galla, 2014**), where $\zeta(t)$ is a rapidly fluctuating random term. The CLE is more computationally efficient at large system sizes, because it does not keep track of every reaction, as they are effectively bundled into time steps.

$$\frac{dm}{dt} = \Omega(G(p(t-\tau)) - S(r)m(t)) + \Omega^{1/2}\zeta_m(t), \tag{16}$$

$$\frac{dp}{dt} = \Omega(F(r)m(t) - \frac{\ln(2)}{\mu_p}p(t)) + \Omega^{1/2}\zeta_p(t), \tag{17}$$

$$\frac{dr}{dt} = \Omega(G_r(p(t)) - \frac{\ln(2)}{\mu_r}r(t)) + \Omega^{1/2}\zeta_r(t), \tag{18}$$

This random term is assumed to have a Gaussian distribution with zero mean, and the fluctuations added at two consecutive time points are uncorrelated (white noise). The variance of the white noise terms are given by

$$\begin{aligned}
\langle\zeta_m(t)\zeta_m(t')\rangle &= (G(p(t-\tau)) + S(r)m)\delta(t-t'), \\
\langle\zeta_p(t)\zeta_p(t')\rangle &= (F(r)m + \frac{\ln(2)}{\mu_p}p)\delta(t-t'), \\
\langle\zeta_r(t)\zeta_r(t')\rangle &= (G_r(p(t)) + \frac{\ln(2)}{\mu_r}r)\delta(t-t'), \\
\langle\zeta_m(t)\zeta_p(t')\rangle &= \langle\zeta_p(t)\zeta_m(t')\rangle = 0, \\
\langle\zeta_m(t)\zeta_r(t')\rangle &= \langle\zeta_r(t)\zeta_m(t')\rangle = 0, \\
\langle\zeta_p(t)\zeta_r(t')\rangle &= \langle\zeta_r(t)\zeta_p(t')\rangle = 0
\end{aligned} \tag{19}$$

The CLE was simulated numerically using an Euler-Maruyama scheme with time steps of 0.1 mins. For history vectors we used a single value repeated over prior time points.

To perform parameter scans, which requires a large number of repeats, we used the Linear Noise Approximation (LNA) because it does not require multiple simulations and is therefore much faster. The LNA is obtained through performing a system size expansion on the CME (**Galla, 2009**). The system size expansion decomposes the time evolution of each species into a deterministic and stochastic contribution, where the relative amplitude of stochastic fluctuations decreases with the inverse of the square root of the system size (i.e. relative fluctuations decrease with higher molecule number)

$$n_i = \phi_i + \Omega^{-1/2}\epsilon_i \tag{20}$$

Where $n_i$ is the number of molecules of species $i$, $\phi_i$ is the concentration of species $i$ as determined by the deterministic rate equations, and $\epsilon_i$ is the random fluctuation with continuous degrees of freedom around the deterministic solution.

After performing the system size expansion, the deterministic equations are recovered when the system size is infinite (no stochasticity). The linear noise approximation (LNA) refers to the next level

of approximation from the system size expansion, valid in the limit of large (but not infinite) system size. The LNA describes stochastic fluctuations around the deterministic trajectory. However, there is a trade-off with accuracy, and the system size expansion can be less accurate than either the dSSA or CLE at low system sizes (*Grima et al., 2011*; *Thomas et al., 2012*). The full detailed description of the system size expansion for the *Hes1*/miR-9 model is shown in Appendix. Even though the CLE and LNA are less accurate than the CME at low system sizes, good agreement between them was observed.

Finally, in addition to the assumptions of the deterministic model (*Goodfellow et al., 2014*), which mostly refer to unknown kinetic parameter values, the stochastic model makes several conceptual assumptions. We assume the chemical system is in well-mixed conditions and the system size is a fixed value. We also assume that the cells are of the same size (volume), and hence the system size parameter is the same within a population of cells and after cell division. When cell division is included in the model, the time of cell division is not constrained and can occur at any point of the HES1 oscillation. Transcription of *Hes1* and miR-9 is controlled by negative feedback and we assume that it is constitutively active when not inhibited. Note that both *Hes1* mRNA and miR-9 are transcribed from DNA, which is assumed to be at constant concentration, and hence both *Hes1* mRNA and miR-9 are produced from the void. In order to make direct comparisons with the previous deterministic description, we do not explicitly model the noise associated with bursting transcription (*Suter et al., 2011*), which is therefore not explicitly distinguished from other sources of inherent noise.

## Comparison of population level simulations with different modes of division

We simulated the time to differentiation in 3000 cells using the CLE, with different system sizes ranging from 0.2 to 50 and looked at the outcome of 50/50 or binomial partitioning at division. The timing distributions using constant or random initial condition models were compared using the Kolmogorov-Smirnov distance, which quantifies the distance between two empirical distributions and can take values between 0 and 1, with lower values indicating higher similarity,

$$KS = \sup_t |F_b(t) - F_e(t)| \tag{21}$$

where $F_b(t)$ is the cumulative distribution function of the timing with binomial initial conditions, and $F_e(t)$ is the cumulative distribution function of the timing with equal 50/50 initial conditions.

## Evaluation of oscillatory dynamics

Central to oscillations is the notion of periodicity. In a stochastic setting, it becomes desirable to be able to distinguish oscillations from random fluctuations, which are dynamic in time but aperiodic. A common method for detecting periodicity is the Fourier transform, which decomposes a given signal into simple sinusoidal oscillating functions. The simple sinusoidal oscillating functions are described by their frequency, and so a Fourier transform takes data in the time domain and represents them in the frequency domain. The Fourier transform is then squared to obtain the power spectrum, and an oscillator is now defined as any process with a peak in its power spectrum at a particular (non-zero) frequency. If the peak is high and sharp, the time series will exhibit high quality oscillations at the dominant frequency with little variability in the peak-to-peak in the time series. However, if the peak in the power spectrum is shallow and broad, there can be significant variation in the peak-to-peak times. When there is no peak in the power spectrum at a non-zero frequency, the process will no longer oscillate and will instead consist of aperiodic fluctuations. A continuum of behaviours is possible, and a suitable metric needs to be defined to measure how sharp the peak in the power spectrum is.

One such measure is the coherence, which is defined as the area under the curve within 20% of the peak frequency as a fraction of the total area under the curve (*Alonso et al., 2007*), as shown in *Figure 7—figure supplement 1*. The coherence is easily calculated, since the power spectrum of a stochastic process can be approximated efficiently using the LNA. Unlike the CLE, the LNA is a linear stochastic differential equation, and it is therefore possible to directly calculate the power spectrum of the stochastic dynamics, even with delay (as described in Appendix).

## Experimental methods

### Cell culture

C17.2 cells (RRID:CVCL_4511)(07062902, Sigma, UK) were grown in DMEM (ThermoFisher Scientific, UK ) with 10% FBS. Primary NS cells were isolated from dissected forebrains of E13.5–15.5 embryos from LUC2:HES1 BAC reporter mice (RRID:IMSR_RBRC06013) and cultured as previously described (*Pollard, 2013*). LUC2:HES1 BAC reporter mice were provided by the RIKEN BRC through the National Bio-Resource Project of the MEXT, Japan (*Imayoshi et al., 2013*). Tau-GFP NS cells were generated from ES cells expressing GFP from the *Mapt* (*tau*) locus (*Bibel et al., 2004*) as previously described (*Conti et al., 2005*). Tau-GFP ES and NS-E (NS derived from E14 ES cells) cells were a gift from Jennifer Nichols (Cambridge Stem Cell Institute, UK). All reported cell lines tested negative for mycoplasma.

### Cell transfection, differentiation and generation of lentivirus reporter cell lines

C17.2 cells were transfected with Lipofectamine 2000 (ThermoFisher Scientific, UK) and Tau-GFP NS cells with Lipofectamine 3000 (ThermoFisher Scientific, UK) as per manufacturers' instructions. 24 hr following transfection, C17.2 cells were differentiated by culture in DMEM with 0.2% FBS (*Lundqvist et al., 2013*) whereas Tau-GFP NS were differentiated towards neurons by gradual removal of the EGF and FGF2 growth factors (*Pollard, 2013*). miRVana miR-9 mimic and control mimic were obtained from ThermoFisher Scientific, UK. pDsRed-miR9 Sensor was a gift from Lynn Hudson (Addgene plasmid # 22742) (*Lau et al., 2008*) and control sensor was pDsRed (Clontech, France). pEGFP N1 (Clontech, France) was used as a transfection control plasmid. To generate the VENUS:HES1 reporter constructs the mouse *Hes1* coding (Accession Number NM008235) and 3'UTR sequence was synthesised with flanking Gateway® att recombination sites and inserted into pUC57 vector, resulting in the pUC57-*Hes1* vector. The *Hes1* gene was then transferred to the 3rd generation 'pLNT-Venus:#' lentiviral transfer by LR clonase recombination reaction (*Bagnall et al., 2015*). The resultant vector is termed pLNT *UbC*-VENUS:HES1 and allows for constitutive expression of N-terminally fused HES1 using Ubiquitin-ligase C promoter (UbC). Additionally, the UbC promoter was replaced with a 2.7 kb sequence of the *Hes1* promoter (upstream of the start codon) using Pac1 and Nhe1 restriction enzymes; the derived vector is termed pLNT 2.7 kb-*Hes1pr*-VENUS: HES1. Lentivirus production and transduction protocols were performed as described before (*Bagnall et al., 2015*). NS-E -*UbC*-VENUS:HES1 cells were FACS sorted on BD Influx Cell Sorter (BD Biosciences, UK) to maximize the VENUS:HES1 positive population.

### qRT-PCR

For synchronised population qRT-PCR, C17.2 cells were serum-starved for 24 hr. Following activation with 10% FBS and at time-points of 30 min, cells were lysed and RNA extracted with TRizol (ThermoFisher Scientific, UK). cDNA was prepared using Superscript III (Invitrogen, UK) as per manufacturers' instructions and qRT-PCR for *Hes1* and *Gapdh* was performed with Taqman gene expression assays (*Hes1* - Mm01342805 m1, *Gapdh*-Mm03302249 g1, ThermoFisher, Scientific, UK). In order to obtain an absolute measurement of miR-9 copy numbers we compared miR-9 content of total RNA extractions of C17.2 cells with a standard curve of known miR-9 dilutions. The standard curve was generated using the following synthetic mature miR-9 sequences (Eurogentec); 23nt UC UUUGGUUAUCUAGCUGUAUGA, 22 nt UCUUUGGUUAUCUAGCUGUAUG, 21 nt UCUUUGGUUA UCUAGCUGUAU, 20 nt UCUUUGGUUAUCUAGCUGUA, 19 nt UCUUUGGUUAUCUAGCUGU (in order of abundance according to miRBase). Synthetic microRNA was 10X serially diluted to enable a standard curve of $1\times10^5 - 10$ copies (5 data points). C17.2 cells were grown in four T25 cm$^2$ tissue culture flasks (Corning, UK) in DMEM with 10% FBS to approx. 80% confluency ( $6\times10^5$ cells). One flask was trypsinised (Trypsin EDTA Sigma, UK) and total cell number counted. The remaining flasks were lysed and total RNA extracted using Ambion miR-Vana kit (AM1561, ThermoFisher, UK) as per manufacturers' instructions. The efficiency of RNA extraction was calculated as 67% and this was incorporated into subsequent copy number calculations. The standard curve samples and 50ng of each total RNA sample were reverse transcribed with the Applied Biosystems TaqMan microRNA Reverse Transcription kit (4366596) as per the manufacturers' protocol. The primers used for reverse transcription and Taqman qRT-PCR were the stem-loop primer set for mouse miR-9-5p (product

4427975, assay ID 001089, ThermoFisher, UK). TaqMan qRT-PCR was performed using TaqMan Fast Advanced master mix reagent (Applied Biosystems 4444557) on Applied Biosystems Step-ONE plus real time PCR system.

## Bioluminescence imaging

200000 primary LUC2:HES1 NS cells up to passage 15 were plated on laminin (Sigma, UK) coated 35 mm glass-bottom dishes (Greiner-Bio One) and imaged in NS proliferation media containing 1 mM D-luciferin (Promega, UK) using a 40x oil objective on an Olympus UK LV200 inverted bioluminescence microscope. 10-minute exposure and 2x2 binning were used. Dishes were maintained at 37°C in 5% $CO_2$. Bioluminescent movies were analysed on Imaris (RRID:SCR_007370)(Version 7.2.2, Bitplane). Images were first subjected to a 3x3 median filter to remove bright spot artefacts from cosmic rays, then individual cells were tracked manually using the 'spots' function.

## Single-molecule Fluorescent In Situ Hybridization (smFISH) and immunofluorescence

A set of 29 Quasar 570 5' end labeled 20 nt smFISH probes were designed against *Hes1* exonic sequence, and 38 Quasar 670 5' end labeled 18 nt smFISH probes designed against *Hes1* intronic sequence (Biosearch Technologies, US). Tau-GFP NS cells were grown on coverslips in NS cell media (*Pollard, 2013*) for 72 hr at 37°C with 5% $CO_2$. Coverslips were washed with 1XPBS, fixed for 20 min with 4% formaldehyde in 1XPBS, washed with 1XPBS, and permeabilized in 70% EtOH at 4°C for 1 hr. Cells were then washed for 5 min in 2XSSC+10% deionized formamide. smFISH probes were diluted to 100 nM in 200 ul hybridization buffer per coverslip (2XSSC, 10% deionized formamide, 10% dextran sulphate). Probes were hybridized for 5 hr in a humidified chamber at 37°C, then cells washed in 2XSSC+10% deionized formamide at 37°C for 30 min. Immunofluorescence was performed post hybridization, incubating with primary antibodies at 4°C for 15 hr, and with secondary antibody at room temperature for 1 hr. Coverslips were mounted using ProLong diamond antifade mountant with DAPI (P36962, ThermoFisher Scientific, UK). Primary antibodies: anti-Tuj1 (RRID:AB_444319)(ab18207, Abcam, UK), rabbit anti-GFP (RRID:AB_10073917)(A11122, ThermoFisher Scientific, UK) and rabbit anti-Aurora-B kinase (RRID:AB_302923)(ab2254 Abcam, UK). Secondary antibody: goat anti-rabbit Alexa Fluor 488 (RRID:AB_143165)(A11008, ThermoFisher Scientific, UK).

## smFISH imaging and analysis

Z-stack images were acquired at 0.2 µm z-increments by Delta Vision widefield microscopy (Olympus IX71 microscope body, Coolsnap HQ CCD camera (Photometrics, US), API XYZ stage), using a 60X (1.42NA) objective. Image stacks were viewed using Imaris (Version 7.2.2, Bitplane), and cell areas segmented manually using background auto-fluorescence in the Quasar 570 channel on a maximum projection of the image. *Hes1* mRNA spots per cell were counted by a semi-automated method, using the inbuilt 'analyze spots' function in Imaris. The following settings were used: estimated XY diameter = 0.3 µM, estimated Z diameter = 0.6 µM, background subtraction = yes, filter type = maximum intensity. The threshold cutoff between signal and background was placed manually at the 'trough' in the spots profile for the maximum intensity filter. Active transcription sites were identified manually, defined as *Hes1* intronic smFISH spots within the nucleus that colocalised with exonic *Hes1* smFISH spots.

## Flow cytometry

Tau-GFP NS cells were harvested at 96 hr, 120 hr, 144 hr and 168 hr of neuronal differentiation following transfection with 40 nM of miR-9 mimic or control mimic. Cells were dissociated using accutase (Sigma, UK) and washed in DMEM-F12 supplemented with 0.2% BSA. Flow cytometry analysis for GFP detection was performed on LSR Fortessa (BD Biosciences, UK) and data were analysed with the FlowJo software (RRID:SCR_008520)(TreeStar, US).

## Fluorescence Correlation Spectroscopy (FCS) protein quantitation

FCS was used to count the number of fluorescent protein molecules expressed in live NS-E cells that were transduced with lentiviral VENUS:HES1 constructs. FCS involves laser excitation and data collection in a tightly focused microscopic region known as the confocal volume. Fluorescent particles

diffusing in the confocal volume give rise to fluorescent intensity fluctuations $F(t)$ (example shown in *Figure 2—figure supplement 1A*) which are analysed over lag time $\tau > 0$ to produce the normalised autocorrelation function (*Figure 2—figure supplement 1B*):

$$R(\tau) = \frac{\langle F(t+\tau)F(t)\rangle}{\langle F(t)\rangle^2} - 1 \tag{22}$$

The amplitude of the autocorrelation function is proportional to 1/c where c represents the number of molecules detected in the confocal volume, i.e. molecule concentration. The shape of $R(\tau)$ can be explained by models taking into consideration multiple species of fluorescent particles characterised by different diffusion constants. Preliminary model selection indicated that two components were required to explain the variability in our data (results not shown) and a non-fluorescent component was added to account for transitions of fluorescent particles to a triplet state. Normalised autocorrelations were fitted with a two component model with triplet state previously used for live cells (*Dross et al., 2009*; *Bagnall et al., 2015*).

FCS experiments were carried out using a Zeiss LSM880 microscope with a Plan-Apochromat 40x, 1.4 NA oil-immersion objective on cells cultured and maintained at 37°C and 5% CO2. Venus (EYFP) fluorescence was excited with 514 nm laser light and emission collected between 516 and 570 nm. Data from individual cell nuclei was collected using 5 x 5 s runs at 0.2 to 0.3% laser power for each measurement using Zen 2.1 (RRID:SCR_013672)(Zeiss) software. We used custom Matlab (RRID:SCR_001622) routines to fit the autocorrelation data (*Figure 2—figure supplement 1B*) and obtain protein concentration in the confocal volume. Measurements collected from cells with count per molecule lower than 500 kHz were excluded from the final results. Single-cell data of number of molecules in the cell nucleus was obtained by adjusting concentration to the average volumetric ratio between nuclear volume and confocal volume. NS-E cells have a nuclear volume of mean 721 fL ± 105 fL (S.D.) estimated using nuclear staining and 3D reconstruction from z-stack images in Imaris. The confocal volume had been previously determined with mean 0.57fL ± 11 fL (S.D.) (*Bagnall et al., 2015*).

## Western blotting

Nuclear extracts of *UbC*-VENUS:HES1 NS-E cells treated with DMSO or MG132 (MerckMillipore, UK) for 3 hr were generated as described in *Schreiber et al. (1989)*. Western blots were performed using 10% NUPAGEBis-Tris gels (ThermoFisher, UK) and Whatman Protran nitrocellulose membrane (Sigma, UK) according to manufacturer's instructions. Antibodies used were anti-HES1 (RRID:AB_590682)(D134-3, MBL, Japan), anti-LAMIN B1 (RRID:AB_443298)(ab16048, Abcam, UK), HRP-linked anti-rat IgG (RRID:AB_10694715)(7077, CST, UK) and HRP-linked anti-rabbit IgG (RRID:AB_2099233) (7074, CST, UK).

## Statistical analysis

Data were analysed using pairwise hypothesis testing methods and multiple hypothesis tests as indicated in the figure legends using Prism 7 (RRID:SCR_002798 )(GraphPad, U.S.A). Statistical significance is reported for p-values <0.05 (*), <0.01(**) and <0.001 (***). Errors are reported as standard error of the mean (S.E.M) calculated from pooled data taking into account standard deviation (S.D) and sample size (n) as S.E.M=S.D./sqrt(n).

## Acknowledgements

This work was supported by a Wellcome Trust Senior Research Fellowship to NP (090868/Z/09/Z), a Sir Henry Wellcome Fellowship to CM (103986/Z/14/Z), a Wellcome Trust Institutional Strategic Support Award (097820/Z/11/B) and a BBSRC Doctoral Training Centre in Systems Biology studentship to NEP. PP holds a BBSRC David Phillips Research Fellowship (BB/I017976/1). MG gratefully acknowledges the financial support of the EPSRC via grant EP/N014391/1. The contribution of MG was generously supported by a Wellcome Trust Institutional Strategic Support Award (WT105618MA). J Boyd was funded by MRC grant MR/K015885/1. DS and MW's work is funded by an MRC grant MR/K015885/1 and a BBSRC grant BB/K003097/1. The authors would also like to thank the Biological Services Facility (BSF), the Bioimaging and Flow Cytometry Facilities of the

University of Manchester for technical support, in particular to Dr Gareth Howell for expert advice on challenging FACS sorts. We thank Dr. Ximena Soto for advice and discussions and Dr Angelica Santiago-Gomez from Robert B Clarke's group at the MCRC, Manchester for technical support and advice with western blotting. The funders had no role in study design, data collection and analysis, decision to publish, or preparation of the manuscript.

## Additional information

### Funding

| Funder | Grant reference number | Author |
| --- | --- | --- |
| Biotechnology and Biological Sciences Research Council | Doctoral Training Centre in Systems Biology | Nick E Phillips |
| Wellcome Trust | Sir Henry Wellcome Postdoctoral Fellowship. 103986/Z/14/Z | Cerys S Manning |
| Wellcome Trust | WT ISSF, Biomedical research Consortia grant, 097820/Z/11/B | Tom Pettini |
| Medical Research Council | MR/K015885/1 | James Boyd<br>David G Spiller<br>Michael RH White |
| Biotechnology and Biological Sciences Research Council | David Phillips Fellowship, BB/I017976/1 | Pawel Paszek |
| Biotechnology and Biological Sciences Research Council | BB/K003097/1 | David G Spiller<br>Michael RH White |
| Engineering and Physical Sciences Research Council | EP/N014391/1 | Marc Goodfellow |
| Wellcome Trust | WT105618MA | Marc Goodfellow |
| Wellcome Trust | 090868/Z/09/Z | Nancy Papalopulu |

The funders had no role in study design, data collection and interpretation, or the decision to submit the work for publication.

### Author contributions
NEP, CSM, TP, VB, EM, NP, Conception and design, Acquisition of data, Analysis and interpretation of data, Drafting or revising the article; PS, MR, Conception and design, Acquisition of data, Analysis and interpretation of data; JBo, Acquisition of data, Analysis and interpretation of data; JBa, PP, Analysis and interpretation of data, Contributed unpublished essential data or reagents; DGS, provided FCS training, designed experiments and had supervisory input into the FCS work, Analysis and interpretation of data; MRHW, supervisory input into the FCS work, Analysis and interpretation of data; MG, TG, Analysis and interpretation of data, Drafting or revising the article

### Author ORCIDs
Michael RH White, http://orcid.org/0000-0002-3617-3232
Marc Goodfellow, http://orcid.org/0000-0002-7282-7280
Nancy Papalopulu, http://orcid.org/0000-0001-6992-6870

### Ethics
Animal experimentation: All animal work was performed under regulations set out by the UK Home Office Legislation under the 1986 United Kingdom Animal Scientific Procedures Act.

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

## Appendix

## Model parameters

**Appendix 1—table 1.** Parameters of the model and their values.

| Parameter | Set 1 | Set 2 | Interpretation | Reference |
|---|---|---|---|---|
| $\tau$ | 29 min | 29 min | Time delay in HES1 protein production | (**Lewis, 2003**) |
| $\alpha_m$ | 1 min$^{-1}$ | 1 min$^{-1}$ | *Hes1* transcription rate in absence of Hes1 protein | |
| $\alpha_p$ | 1 min$^{-1}$ | 2 min$^{-1}$ | Translation rate | |
| $\alpha_r$ | 1 min$^{-1}$ | 1 min$^{-1}$ | miR-9 transcription rate in absence of Hes1 protein | |
| $p_0$ | 390 | 390 | Amount of protein required to reduce *Hes1* transcription by half | Fitted in (**Goodfellow et al., 2014**) |
| $n_0$ | 5 | 5 | Quantifies the step-like nature of $G$ | (**Monk, 2003**) |
| $r_0$ | 100 | 80 | Amount of miR-9 required to reduce *Hes1* degradation by half | |
| $m_0$ | 5 | 5 | Quantifies the step-like nature of $S$ | |
| $b_l$ | $\ln(2)/20$min$^{-1}$ | $\ln(2)/20$ min$^{-1}$ | Lower bound for *Hes1* mRNA half-life | (**Bonev et al., 2012**) |
| $b_u$ | $\ln(2)/35$min$^{-1}$ | $\ln(2)/35$ min$^{-1}$ | Upper bound for *Hes1* mRNA half-life | (**Bonev et al., 2012**) |
| $r_1$ | 300 | 100 | Amount of miR-9 required to reduce HES1 protein translation by half | |
| $m_1$ | 5 | 5 | Quantifies the step-like nature of $F$ | |
| $n_1$ | 5 | 5 | Quantifies the step-like nature of $G_r$ | |
| $\mu_p$ | 22 min | 22 min | Half-life of HES1 protein | (**Hirata et al., 2002**) |
| $p_1$ | 260 | 280 | Amount of protein required to reduce miR-9 production rate by half | |
| $\mu_r$ | 1000 min | 1000 min | Half-life of miR-9 | |

## Deterministic dynamics

Starting with the microscopic dynamics of the *Hes1*/miR-9 described in the Materials and Methods, we apply the system size expansion as described in Brett and Galla (**Brett and Galla, 2013**) and Galla (**Galla, 2009**). To lowest order in $\Omega^{-1/2}$, one recovers the deterministic dynamics described in Goodfellow et al. (**Goodfellow et al., 2014**)

$$\frac{dm}{dt} = G(p(t-\tau)) - S(r)m, \tag{23}$$

$$\frac{dp}{dt} = F(r)m - \frac{\ln(2)}{\mu_p}p, \tag{24}$$

$$\frac{dr}{dt} = G_r(p) - \frac{\ln(2)}{\mu_r}r. \tag{25}$$

## Linear noise approximation

The next-to-leading order in $\Omega^{-1/2}$ represents the LNA and recovers the probability distribution describing random deviations in molecule numbers around the deterministic trajectory. The fluctuations in mRNA, protein and miR-9 molecule number are denoted $\epsilon_m$, $\epsilon_p$ and $\epsilon_r$, respectively. In order to account for noise induced dynamics below the Hopf bifurcation, we assume that the deterministic dynamics reach a fixed-point. When the deterministic dynamics have reached a fixed-point (i.e. after large time), the dynamics of the fluctuations can be approximated by the linear Langevin equations

$$\frac{d\epsilon_m}{dt} = G'(p^*)\epsilon_p(t-\tau) - S(r^*)\epsilon_m(t) - S'(r^*)m^*\epsilon_r(t) + \zeta_m(t), \tag{26}$$

$$\frac{d\epsilon_p}{dt} = F(r^*)\epsilon_m(t) + F'(r^*)m^*\epsilon_r(t) - \frac{\ln(2)}{\mu_p}\epsilon_p(t) + \zeta_p(t), \tag{27}$$

$$\frac{d\epsilon_r}{dt} = G'_r(p^*)\epsilon_p(t) - \frac{\ln(2)}{\mu_r}\epsilon_r(t) + \zeta_r(t), \tag{28}$$

where $G'(p^*)$, $S'(r^*)$, $F'(r^*)$ and $G'_r(p^*)$ represent are linearisations of the Hill functions $G, S$ and $G_r$ around the deterministic fixed point, and where $\zeta(t)$ represents white noise. Within the linear-noise approximation this noise is assumed to be additive. At the deterministic fixed point, the noise correlators are given by

$$\begin{aligned}
\langle \zeta_m(t)\zeta_m(t') \rangle &= (G(p^*) + S(r^*)m^*)\delta(t-t'), \\
\langle \zeta_p(t)\zeta_p(t') \rangle &= (F(r^*)m^* + \frac{\ln(2)}{\mu_p}p^*)\delta(t-t'), \\
\langle \zeta_r(t)\zeta_r(t') \rangle &= (G_r(p^*) + \frac{\ln(2)}{\mu_r}r^*)\delta(t-t'), \\
\langle \zeta_m(t)\zeta_p(t') \rangle &= \langle \zeta_p(t)(\zeta_m(t') \rangle = 0, \\
\langle \zeta_m(t)\zeta_r(t') \rangle &= \langle \zeta_r(t)\zeta_m(t') \rangle = 0, \\
\langle \zeta_p(t)\zeta_r(t') \rangle &= \langle \zeta_r(t)\zeta_p(t') \rangle = 0
\end{aligned} \tag{29}$$

The LNA therefore describes a process with linear delay relaxation dynamics supplemented by additive noise. This constitutes a multivariate Ornstein-Uhlenbeck process (with delay). To analyse this further, we carry out a Fourier transform (with respect to time), and obtain

$$i\omega\tilde{\epsilon}_m(\omega) = G'(p^*)\tilde{\epsilon}_p(\omega)e^{-i\omega t} - S(r^*)\tilde{\epsilon}_m(\omega) - S'(r^*)m^*\tilde{\epsilon}_r(\omega) + \tilde{\zeta}_m(\omega), \tag{30}$$

$$i\omega\tilde{\epsilon}_p(\omega) = F(r^*)\tilde{\epsilon}_m(\omega) + F'(r^*)m^*\tilde{\epsilon}_r(\omega) - \frac{\ln(2)}{\mu_p}\tilde{\epsilon}_p(\omega) + \tilde{\zeta}_p(\omega), \tag{31}$$

$$i\omega\tilde{\epsilon}_r(\omega) = G'_r(p^*)\tilde{\epsilon}_p(\omega) - \frac{\ln(2)}{\mu_r}\tilde{\epsilon}_r(\omega) + \tilde{\zeta}_r(\omega). \tag{32}$$

In matrix form this can be represented as

$$\underline{M}(\omega)\tilde{\boldsymbol{\epsilon}}(\omega) = \tilde{\boldsymbol{\zeta}}(\omega), \tag{33}$$

where

$$\underline{M}(\omega) = \begin{pmatrix} i\omega + S(r^*) & -G'(p^*)e^{-i\omega t} & S'(r^*)m^* \\ -F(r^*) & i\omega + \frac{\ln(2)}{\mu_p} & -F'(r^*)m^* \\ 0 & -G'_r(p^*) & i\omega + \frac{\ln(2)}{\mu_r} \end{pmatrix}. \tag{34}$$

It is now useful to introduce the matrix of spectra, $\underline{S}(\omega)$, with $S_{ij}(\omega) = \langle \tilde{\epsilon}_i(\omega)\tilde{\epsilon}_j(\omega)^\dagger \rangle$, where $i,j$ represent the variables $m, p$ and $r$ above. One then has

$$\underline{S}(\omega) = \underline{M}(\omega)^{-1} \langle \tilde{\zeta}(\omega)\tilde{\zeta}(\omega)^\dagger \rangle \underline{M}(\omega)^{\dagger-1},$$ (35)

where $\langle \tilde{\zeta}(\omega)\tilde{\zeta}(\omega)^\dagger \rangle$ is

$$\langle \tilde{\zeta}(\omega)\tilde{\zeta}(\omega')^\dagger \rangle = \begin{pmatrix} (G(p^*) + S(r^*)m^*) & 0 & 0 \\ 0 & (F(r^*)m^* + \frac{\ln(2)}{\mu_p}p^*) & 0 \\ 0 & 0 & (G_r(p^*) + \frac{\ln(2)}{\mu_r}r^*) \end{pmatrix} \delta(\omega + \omega').$$ (36)

The further analysis is then based on a numerical evaluation of **Equation 35**.

