## [Decision Letter]

Thank you for submitting your article "Stochasticity in the miR/Hes1 oscillatory network can account for clonal heterogeneity in the timing of differentiation" for consideration by *eLife*. Your article has been reviewed by three peer reviewers, one of whom, Alejandro Sánchez Alvarado, is a member of our Board of Reviewing Editors and the evaluation has been overseen by Aviv Regev as the Senior Editor. The following individuals involved in review of your submission have agreed to reveal their identity: William A Harris (Reviewer #2).

The reviewers have discussed the reviews with one another and the Reviewing Editor has drafted this decision to help you prepare a revised submission.

Summary:

Timing in biological systems is a fundamental topic that has a long history, and is enjoying resurgence due to improvements in microscopy and theory. Phillips et al. computationally model the effects of intrinsic noise on the Hes1/miR-9 oscillator, and propose that the stochasticity generated by such noise is primarily caused by low molecular numbers of interacting species. The model offers a number of advantages to understand cell differentiation over the more widely accepted deterministic model. In this proposed stochastic/probabilistic model, the authors show that increased stochasticity spreads the timing of differentiation in a population, which potentially can account for clonal heterogeneity in a developing population of cells. This is a feature of real development that is not inherent in deterministic models of this network in which when such equivalent cells are modelled using differential equations, as in such models all cells differentiate at the same time. The authors also show that such stochastic models can add robustness to the oscillations by causing cells that might have differentiated due to near threshold levels of Hes1 to remain oscillating. The paper thus gives a detailed mathematical potential explanation for recent observations of clonal variability. It suggest that intrinsic stochasticity based of small numbers of interacting molecules is a basis for the kind of variable output in the timing of differentiation in neural progenitors.

Essential revisions:

The following issues need to be addressed fully in a revised submission.

1) The manuscript is at times confusing when it uses the different models. In 3.1 we are given a very gentle introduction to noise in molecular systems, but we are told nothing about the CME, dSSA and CLE. The clarity of the manuscript would benefit if each of these methods were given a one-two sentence introduction, whether in 3.1, or when they first appear so that the reader can understand the logic of the approach. For example, it is critical when deterministic and stochastic methods are compared that the reader understands what it means for them to have the same parameters. We realize the methods and supplements contain this information and there, it is well-written and solid, but others may not read these closely because of the mathematics and for them it may be opaque.

2) The proposed model is just one of several possible explanations that can account for the variability in clone sizes or differentiation statistics. The authors might want to discuss these other models in more detail, particularly their comparative strengths and weaknesses. For example, the authors show that if a progenitor divides such that the two daughters inherit different proportions of the parent cell's small number of Hes1 molecules, this too can drive stochastic differentiation timings even when a deterministic model based on differential equations is applied. Another recently published model (Agarwal et al. 2016 in PLoS One) shows that the variability of nuclear position in neural progenitors, which are thought to have a gradient of intracellular Notch ICD, is sufficient to explain the stochastic spread in differentiation seen in the zebrafish retina. There are other models that can inject stochasticity into these systems, either intrinsically (e.g., things that change the cell cycle) or extrinsically (based on interactions between cells).

3) The authors show that increasing miR-9 promotes neuronal differentiation by counting percentage of cells expressing Tuj1 or Tau-GFP in population level (Figure 1). However, the real distribution of time-to-differentiation at the single cell level via direct experimental measurement is missing. Since this is one of the key advancements of the study, showing the live image of dynamic gene expression (Hes1) and the distribution of timing of cell fate transition during normal differentiation, at least in the cultured cell lines used, seems necessary. Therefore, it would make the paper much stronger if the authors could somehow test the model (or some key aspect of it) experimentally. If live imaging is not feasible, another approach, for example, might be to focus on a key parameter of the model, called omega (based on the concentration of molecules within a given volume). Perhaps one could predict omega in a particular cell type because the distribution of oscillations and differentiation times works well if the value of omega is within a certain range and then test what exactly omega is in these cells. Is it close to the predicted value? Or perhaps they could experimentally change the starting omega conditions (quantitatively change the amount of miR9 or Hes1) in a population of cells and see if if gives the predicted outcomes.

4) How the results in Figure 1 are directly relevant to this paper is not clear. Do they add something that was not clear from previous work and/or are required to motivate the examination of stochastic effects?

5) Single cell trace in Figure 2: we are shown one cell, but cannot ascertain if this is typical. What was the culture density? Was this cell in contact with other cells in culture? What is the system being modeled? The cell is used to illustrate the point that if you use a deterministic model it resembles biological data more poorly that a stochastic model. As such, this is a trivial statement. For single cell data to meaningfully inform the discussion or contribute to our understanding there would have to be some kind of quantification and analysis of the data of a population of cells to say whether it fit a particular stochastic description that could be used to argue about the properties of the system. Ideally, one would compare two competing stochastic descriptions to determine which was more useful.

6) We also wonder how much fitting has had to be done to get these stochastic models to work well. There are a number of parameters and starting conditions that are used. Can any of these be fixed by experimental observations? The more unfixed parameters and starting conditions there are, the weaker the model is.

7) In the Results section, the authors claim that "spontaneously differentiated cells were mostly negative for Hes1 mRNA transcript." However, of the seven cells outlined in Figure 1, almost all of them are showing GFP signal if one takes a closer look at them. To support their conclusion that Hes1 expression is switched off upon differentiation, the authors need to characterize the differentiation of these cells using more specific antibodies or functional assay. Also, quantitative and time-series measurements of the abundance of hes1 and miR-9 expression, along with the tau-GFP appearance in single cell level will help to demonstrate a direct relationship.

8) The power spectrum technique is strong and appears to be a novel contribution, yet it is well-hidden. It is an on-going problem a noisy time series to answer the question whether it comes from an oscillator or a fluctuating, but non-oscillating system. It would therefore be helpful to introduce this in a clearer manner, and show how it works on the deterministic case first.

9) Figure 7 and the text describing it are difficult to follow. Are we right in thinking this is essentially describing a negative result? It might be better to describe this the other way around – first to say what is the effect of 50-50 distribution at division for a deterministic model, and how this is perturbed by a binomial distribution. Then, show how the stochastic model is "protected" against the effect of binomial distribution.

10) We are somewhat confused by the distinction the authors make about the "advantageous" use of biological noise. According to the narrative, stochasticity introduces a variation in the differentiation time due to the miR timing mechanism (which is good), but then stochasticity reduces the variation in timing due to cell division (which apparently is also good). Cannot one say that the total variation in timing is influenced by the effect of stochasticity on both the miR timing circuit and distribution at division?

11) Overall, the data set and data analysis are of high quality and should be of great interest to the readership of *eLife*. The one shortcoming of this article, which is acknowledged by the senior author, is a lack of experimental evidence to support key corollaries of the model. The use of prior data and some new experiments to hone and refine the model is laudable and make this article a rigorous one. However, putting forward experimental evidence to support the functional importance of the shift of the timing of differentiation on clonal heterogeneity, e.g., heterogeneous expression of cell markers, would have been welcomed. In its present form, I remain unconvinced about the spread distribution of time-to-differentiation is caused by decreased system size (molecular copy number) in the real developmental context.

---

## [Author Response]

*1) The manuscript is at times confusing when it uses the different models. In 3.1 we are given a very gentle introduction to noise in molecular systems, but we are told nothing about the CME, dSSA and CLE. The clarity of the manuscript would benefit if each of these methods were given a one-two sentence introduction, whether in 3.1, or when they first appear so that the reader can understand the logic of the approach. For example, it is critical when deterministic and stochastic methods are compared that the reader understands what it means for them to have the same parameters. We realize the methods and supplements contain this information and there, it is well-written and solid, but others may not read these closely because of the mathematics and for them it may be opaque.*

We apologise for this opaqueness, which was due to space limitations. We have re-written this section, adding general information about the mathematical methods into the main body of the paper (Results section). Some sentences were moved forward from the Materials and methods section (with the Journal’s permission) to increase the word count of the main body of the paper. We hope that this will help to making all parts of the manuscript accessible to the experimentalist as well as the theoretician.

*2) The proposed model is just one of several possible explanations that can account for the variability in clone sizes or differentiation statistics. The authors might want to discuss these other models in more detail, particularly their comparative strengths and weaknesses. For example, the authors show that if a progenitor divides such that the two daughters inherit different proportions of the parent cell's small number of Hes1 molecules, this too can drive stochastic differentiation timings even when a deterministic model based on differential equations is applied. Another recently published model (Agarwal et al. 2016 in PLoS One) shows that the variability of nuclear position in neural progenitors, which are thought to have a gradient of intracellular Notch ICD, is sufficient to explain the stochastic spread in differentiation seen in the zebrafish retina. There are other models that can inject stochasticity into these systems, either intrinsically (e.g., things that change the cell cycle) or extrinsically (based on interactions between cells).*

We have done experiments to strengthen the proposed model (described below) but we agree that there are other possible explanations, which may not be mutually exclusive. Thank you for the suggestion of the Aqgarwal et al. paper. We note that this paper was purely theoretical i.e. no experimental evidence was provided for the proposed steep NICD intracellular gradient. The Agqarwal et al. paper also assumes that Hes1 only exists in an ON or OFF state, so it ignores the evidence that Hes1 oscillates. Nevertheless, some variant of their model may apply; we have added this reference. We also discuss noise introduced by asymmetries at cell division (Huh and Paulson, 2011) and the effect of noise in the Notch pathway (Jenkins, 2015). We note that future work can extend our model to multicellular systems. Please let us know if we are missing other relevant references and we will gladly add them.

*3) The authors show that increasing miR-9 promotes neuronal differentiation by counting percentage of cells expressing Tuj1 or Tau-GFP in population level (Figure 1). However, the real distribution of time-to-differentiation at the single cell level via direct experimental measurement is missing. Since this is one of the key advancements of the study, showing the live image of dynamic gene expression (Hes1) and the distribution of timing of cell fate transition during normal differentiation, at least in the cultured cell lines used, seems necessary. Therefore, it would make the paper much stronger if the authors could somehow test the model (or some key aspect of it) experimentally. If live imaging is not feasible, another approach, for example, might be to focus on a key parameter of the model, called omega (based on the concentration of molecules within a given volume). Perhaps one could predict omega in a particular cell type because the distribution of oscillations and differentiation times works well if the value of omega is within a certain range and then test what exactly omega is in these cells. Is it close to the predicted value? Or perhaps they could experimentally change the starting omega conditions (quantitatively change the amount of miR9 or Hes1) in a population of cells and see if if gives the predicted outcomes.*

We have taken this suggestion on board and we have added new experimental data to support our model. As suggested, we have focused on a key parameter of the model, which is the starting concentration of miR-9. We have done a computational experiment to increase the starting concentration of miR-9 in a population of cells. Alongside this, we have experimentally overexpressed a miR-9 mimic or control mimic in C17.2 neural progenitor cells and Tau-GFP NS cells. We show by several methods (i.e. Tuj1 expression in fixed populations, flow cytometry analysis for GFP expression in Tau-GFP NS cells), that as predicted by our model, overexpression of miR-9 brings neuronal differentiation forward in time. Importantly, we also show that the addition of miR-9 reduces the time distribution to differentiation at the population level, which agrees with the stochastic model and is a novel feature of it. This shows that the stochastic model is able to make predictions that are consistent with experimental measurements. The new results are shown in Figure 6 and Figure 6—figure supplement 1 and Figure 6—figure supplement 2 and are presented in a new section entitled “Increased miR-9 concentration shifts timing and reduces spread of differentiation”.

*4) How the results in Figure 1 are directly relevant to this paper is not clear. Do they add something that was not clear from previous work and/or are required to motivate the examination of stochastic effects?*

We apologise that this figure caused confusion. Part of it was intended to motivate this study and part was to support the model. We have generated a lot more experimental data to motivate and support this study, such that Figure 1 has now been redesigned and split into two. The old Figure 1 has been moved to new Figure 1—figure supplement 1. The old Figure 1 has been moved to Figure 6—figure supplement 1 and old Figure 1 has been omitted as it has been superseded by the flow cytometry analysis shown in Figure 6—figure supplement 2 (New Figure 6 is a new figure with new data addressing major point 3 above). Figure 1 now starts with a description of the smFISH design (Figure 1). See also new text in subsection “A “finite number” stochastically simulated network is justified by experimental data”)

*5) Single cell trace in Figure 2: we are shown one cell, but cannot ascertain if this is typical. What was the culture density? Was this cell in contact with other cells in culture? What is the system being modeled? The cell is used to illustrate the point that if you use a deterministic model it resembles biological data more poorly that a stochastic model. As such, this is a trivial statement. For single cell data to meaningfully inform the discussion or contribute to our understanding there would have to be some kind of quantification and analysis of the data of a population of cells to say whether it fit a particular stochastic description that could be used to argue about the properties of the system. Ideally, one would compare two competing stochastic descriptions to determine which was more useful.*

Yes, the reviewer is correct in saying that Figure 2 was used to illustrate the point that a deterministic model resembles the data more poorly than a stochastic one. We don’t’ think that this is trivial, because the explicit comparison of the stochastic and deterministic models with experimental data has not been made before for the Hes1 system. It will of course be familiar to those working on the Hes1 oscillator from inspection of their own raw data but other oscillators (such as circadian) are often more regular. Nevertheless, we took this point on board and deepened our analysis. We are now including the traces of 15 representative cells in Figure 3—figure supplement 2 and we provide cell culture information in the Materials and methods. We also provide Luciferase imaging snapshots of a time series in Figure 3—figure supplement 1.

Quantification and analysis of the data of a population of cells was already in the figure. We used qRT-PCR as a way to obtain population level data and to show that they fit synthetic population data in the stochastic framework better than in a deterministic one. This is shown in Figure 3.

*6) We also wonder how much fitting has had to be done to get these stochastic models to work well. There are a number of parameters and starting conditions that are used. Can any of these be fixed by experimental observations? The more unfixed parameters and starting conditions there are, the weaker the model is.*

Some of the parameters of the model, such as mRNA and protein degradation rates, have been experimentally measured either from our work or the work of others. Other parameters have not been directly measured, but have been fitted in order to reproduce dynamical behavior. We apologise that this was not clear. We have now improved the table of parameters by including textual description of the terms and also references to previous literature (see Supplementary file Table 1)

*7) In the Results section, the authors claim that "spontaneously differentiated cells were mostly negative for Hes1 mRNA transcript." However, of the seven cells outlined in Figure 1, almost all of them are showing GFP signal if one takes a closer look at them. To support their conclusion that Hes1 expression is switched off upon differentiation, the authors need to characterize the differentiation of these cells using more specific antibodies or functional assay. Also, quantitative and time-series measurements of the abundance of hes1 and miR-9 expression, along with the tau-GFP appearance in single cell level will help to demonstrate a direct relationship.*

To strengthen the data in Figure 1, we have done what the reviewer requested. We quantitated GFP signal intensity following Hes1 smFISH in Tau-GFP NS cells, which allowed us to separate the GFP negative and GFP positive cells based on the intensity of staining. The new Figure 1—figure supplement 3, shows a clear separation of negative and positive populations. The quantitation of Hes1 smFISH (Figure 1) was then done on the GFP negative population (undifferentiated cells). To further support the conclusion that Hes1 is switched off upon differentiation we performed smFISH with an intronic Hes1 probes, which in combination with the Hes1 exonic probes, reveals the sites of transcription. These data showed that Tau-GFP positive cells are negative for Hes1 transcription. This is shown in Figure 1.

To address the issue of abundance we have added new data on the absolute quantification of miR-9 (by stem loop PCR) and HES1 protein (by Fluorescence Correlation Spectroscopy (FCS) and Western Blotting), The absolute copy numbers, expressed as a population average, are well below the global reported median, justifying the finite number approach. These are in Figure 1—figure supplement 4 and Figure 1—figure supplement 5 (miR-9 stem loop qRT-PCR) and in Figure 2 (protein FCS and WB), new section in the Results and discussed in the Discussion.

*8) The power spectrum technique is strong and appears to be a novel contribution, yet it is well-hidden. It is an on-going problem a noisy time series to answer the question whether it comes from an oscillator or a fluctuating, but non-oscillating system. It would therefore be helpful to introduce this in a clearer manner, and show how it works on the deterministic case first.*

We are pleased to see that the reviewers find that this is an ongoing problem, which needs to be addressed. We have developed a novel method for classifying stochastic time series as oscillatory or not but the full method is beyond the scope of this paper. A manuscript describing this method is currently under review in PLOS Comp Biology (“Identifying stochastic oscillations in single cell live imaging time series using Gaussian processes”, by Nick E. Phillips, Cerys Manning, Nancy Papalopulu and Magnus Rattray).

*9) Figure 7 and the text describing it are difficult to follow. Are we right in thinking this is essentially describing a negative result? It might be better to describe this the other way around – first to say what is the effect of 50-50 distribution at division for a deterministic model, and how this is perturbed by a binomial distribution. Then, show how the stochastic model is "protected" against the effect of binomial distribution.*

We agree that this was difficult to follow; we have extensively re-written the text (subsection “Robustness of timing to perturbations from asymmetric inheritance at cell division”) describing these results and we have resigned the figure to start with the 50:50 distribution of the deterministic system, as suggested by the reviewers. This is now new Figure 9 and we think that it is much clearer; thank you for your comment.

*10) We are somewhat confused by the distinction the authors make about the "advantageous" use of biological noise. According to the narrative, stochasticity introduces a variation in the differentiation time due to the miR timing mechanism (which is good), but then stochasticity reduces the variation in timing due to cell division (which apparently is also good). Cannot one say that the total variation in timing is influenced by the effect of stochasticity on both the miR timing circuit and distribution at division?*

Apologies that we presented this in a way that was not clear. We do say that stochasticity introduces a variation in the differentiation time due to the miR-9 timing mechanism and this is good, as the reviewer says, because it allows time for neuronal diversity and feedback control (as we speculate in the Discussion). However, the reviewer has misunderstood the second part. We say that stochasticity does not change the variation in timing due to cell division. In fact, our data show that the effect of “finite number” noise and “partitioning at cell division” noise are not additive. This can be seen by comparing columns D (50:50 division) and E (random division) of the stochastic network, in Figure 9. This why we say that the stochastic system is “robust” to partitioning noise. We have extensively re-written this Result section (subsection Robustness of timing to perturbations from asymmetric inheritance at cell division) as well as the Discussion and we hope this point is now clearer.

*11) Overall, the data set and data analysis are of high quality and should be of great interest to the readership of eLife. The one shortcoming of this article, which is acknowledged by the senior author, is a lack of experimental evidence to support key corollaries of the model. The use of prior data and some new experiments to hone and refine the model is laudable and make this article a rigorous one. However, putting forward experimental evidence to support the functional importance of the shift of the timing of differentiation on clonal heterogeneity, e.g., heterogeneous expression of cell markers, would have been welcomed. In its present form, I remain unconvinced about the spread distribution of time-to-differentiation is caused by decreased system size (molecular copy number) in the real developmental context.*

Thank you for finding our paper of high quality and great interest. We have done several new experiments to strengthen the paper, as detailed in our response to the major and minor points above. In particular, to support the finite number stochastic simulations, in addition to the previous Hes1 mRNA copy number (Figure 1), we are now presenting absolute quantitative data for miR-9 (Figure 1—figure supplement 4 and Figure 1—figure supplement 5) and HES1 protein abundance (Figure 2); both of these support the idea of low molecular copy number. In addition, we have added theoretical and experimental data to show that increasing the initial concentration of miR-9 speeds up differentiation and reduces its timing spread (Figure 6, Figure 6—figure supplement 1 and Figure 6—figure supplement 2).